# Is there anything good about conspiracy beliefs? Belief in COVID-19 conspiracy theories is associated with benefits to well-being

**Javier A. Granados Samayoa[1], Courtney A. Moore[2], Benjamin C. Ruisch[3], Jesse T. Ladanyi[2], Russell H. Fazio[2]\***

**1** Annenberg Public Policy Center, University of Pennsylvania, Philadelphia, Pennsylvania, United States of America, **2** Department of Psychology, The Ohio State University, Columbus, Ohio, United States of America, **3** School of Psychology, University of Kent, Canterbury, United Kingdom

\* fazio.11@osu.edu

## Abstract

Recent theorizing suggests that people gravitate toward conspiracy theories during difficult times because such beliefs promise to alleviate threats to psychological motives. Surprisingly, however, previous research has largely failed to find beneficial intrapersonal effects of endorsing an event conspiracy theory for outcomes like well-being. The current research provides correlational evidence for a link between well-being and an event conspiracy belief by teasing apart this relation from (1) the influence of experiencing turmoil that nudges people toward believing the event conspiracy theory in the first place and (2) conspiracist ideation—the general tendency to engage in conspiratorial thinking. Across two studies we find that, when statistically accounting for the degree of economic turmoil recently experienced and conspiracist ideation, greater belief in COVID-19 conspiracy theories concurrently predicts less stress and longitudinally predicts greater contentment. However, the relation between COVID-19 conspiracy belief and contentment diminishes in size over time. These findings suggest that despite their numerous negative consequences, event conspiracy beliefs are associated with at least temporary intrapersonal benefits.

## Introduction

The public profile of conspiracy theories—explanations for events that involve secretive plots by nefarious and powerful groups of people [1]—has seemingly never been higher. Flat earth conventions, acts of violence inspired by QAnon, and the wide assortment of unsupported explanations surrounding the COVID-19 pandemic have brought conspiracy theories to the forefront of the public's mind. It is not surprising, then, that belief in conspiracy theories appears to align with the experience of turmoil [2]. Consistent with this idea, a recent multi-national study found that people living in countries experiencing more conflict tended to believe COVID-19 conspiracy theories to a greater extent [3]. The current study leverages the impactful real-world situation created by the COVID-19 pandemic to explore the effects

**Data availability statement:** Data, syntax, and a full list of measures for Study 1 are available at https://osf.io/7b3xp/?view_only=f649bb-67327c40629d9b63b3b2188fd5. To access the publicly-available data and the accompanying materials for Study 2, please visit https://osf.io/v2zur/. For syntax specific to the analyses reported in Study 2, see https://osf.io/7b3xp/?view_only=f649bb-67327c40629d9b63b3b2188fd5.

**Funding:** This work was supported by a RAPID grant from the National Science Foundation under Award ID BCS-2031097 (RHF). The funders had no role in study design, data collection and analysis, decision to publish, or preparation of the manuscript.

**Competing interests:** The authors have declared that no competing interests exist.

of believing a conspiracy theory regarding a specific event on subjective well-being. Before outlining the rationale for our hypotheses, we provide a brief review of the conspiracy theory literature.

## A brief review of the literature on conspiracy theories

### Situational predictors of belief in conspiracy theories

In line with the recent findings discussed above, there is a growing body of correlational evidence that experiencing hardship is associated with endorsing conspiracy theories. For instance, personal income predicts belief in conspiracy theories such that the least wealthy endorse conspiracy theories to a greater extent [2]. Moreover, greater country-level economic inequality—as assessed by the Gini coefficient—predicts greater endorsement of a variety of conspiracy theories [4] and greater national corruption predicted stronger endorsement of COVID-19 conspiracy theories [5].

Experimental work further supports these findings. Participants in one set of experiments engaged in an imagination exercise intended to create the perception of high- versus low-income inequality. The results consistently revealed that individuals who were randomly assigned to perceive greater economic inequality endorsed conspiracy theories to a greater extent [4]. In an experiment more directly targeting emotional experience, people in an anxiety-inducing situation (waiting to take an exam) reported greater endorsement of conspiracy theories about Jewish people than did people in a more relaxed setting (simply waiting to hear a lecture; [6]). Other experimental work has found that people report greater belief in conspiracy theories when induced to experience uncertainty, a lack of control, or ostracism [7]. Taken together, the accumulated literature suggests that people gravitate toward believing conspiracy theories when situations threaten motives that are existential (related to feelings of control and security), epistemic (related to a sense of understanding and certainty), or social (related to the desire to view the self and one's social groups in a positive light) in nature [1,8].

### Individual differences in belief in conspiracy theories

As articulated above, situational factors can nudge people toward endorsing conspiracy theories. Above and beyond such influences, however, there are individual differences in the degree to which people believe conspiracy theories. Conspiracist ideation (also referred to as conspiracy mentality; see [8] for a discussion of terminology)—an individual's general tendency to engage in conspiratorial thinking—has been assessed in a variety of ways. What unites most common measures is a focus on assessing agreement with decontextualized ideas about secret plots by powerful groups. Specifically, such measures ask participants to report the extent to which they agree with a number of general statements (e.g., "The government has employed people in secret to assassinate others") that capture the essence underlying many specific real-life conspiracy theories. Such general measures have been shown to predict belief in event conspiracy theories (e.g., [9]; see [10], for a review) and attitudes towards issues that are the subject of conspiracy theories like vaccination and the use of genetically-modified organisms [11].

### On the distinction between conspiracist ideation and belief in a specific event conspiracy theory

When people use the term "conspiracy beliefs," they are sometimes referring to a person's general tendency to believe conspiracy theories—their level of conspiracist ideation—and sometimes to a person's endorsement of a specific conspiracy theory (or theories) about a specific

event. We use the term "event conspiracy belief" to refer to subjective convictions regarding the existence of a conspiracy in relation to a specific event. It must be noted, however, that a general tendency to believe conspiracy theories —is not the same as an *event conspiracy belief* [12]. Certainly, conspiracist ideation is related to the likelihood of belief in more narrow, specific event-based conspiracy theories. Indeed, one would expect a person who generally tends to believe conspiracy theories to be attracted to explanations for specific events that contain conspiratorial content [13]. However, belief in a given specific conspiracy theory is not in itself an indicator of conspiracist ideation. A person who is not particularly high in conspiracist ideation may come to endorse a conspiracy theory as a result of situational forces. For instance, research conducted during the COVID-19 pandemic found that experiencing greater economic hardship due to the pandemic predicted greater belief in COVID-19 conspiracy theories, and understandably, this relation was particularly pronounced among those high in conspiracist ideation. Importantly, however, the association between economic hardship and belief in COVID-19 conspiracy theory was evident even among people relatively *low* in conspiracist ideation [14]. Stated differently, even those who are not prone to conspiratorial thinking can come to believe a specific event conspiracy theory given sufficient situational influence.

## Consequences of conspiracy beliefs

Aside from exploring the reasons why people gravitate toward conspiracy theories, attention also has been devoted to understanding the consequences of holding such beliefs. In their review of the literature, Douglas and colleagues found that both belief in conspiracy theories regarding a given event or issue (e.g., 9/11 or genetically modified foods) and conspiracist ideation are primarily associated with undesirable outcomes [8]. In the realm of health behavior, Black men who reported greater belief in HIV/AIDS-related conspiracy theories (e.g., AIDS was created to control the Black population) also reported more negative attitudes toward condoms and less condom use [15]. Furthermore, parents who endorsed vaccine-related conspiracy theories to a greater extent were less likely to vaccinate their children [16, 17] and those who believed COVID-19 conspiracy theories were less likely to engage in health-protective behaviors [18]. Experimental evidence suggests that exposure to anti-vaccine conspiracy theories lowers vaccination intentions [16].

Importantly, and moving beyond belief in any given conspiracy theory, conspiracist ideation in and of itself also has been linked to negative health consequences: people who reported greater conspiracist ideation during an initial baseline wave of data collection during the early weeks of the COVID-19 pandemic were more likely to report having contracted COVID-19 when re-assessed several months later [19]. In general, then, the endorsement of conspiracy theories, whether it be in the form of a specific event conspiracy belief or the general tendency to engage in conspiratorial thinking, appears to be associated with negative consequences for the believer and those around them.

However, one potential consequence of endorsing conspiracy theories is conspicuous by its absence: there is a surprising lack of evidence that believing a specific event conspiracy theory fulfills the existential, epistemic, and social motives that seem to drive people to believe such beliefs in the first place [1]. That is, even though one reason people gravitate toward conspiracy theories is the experience of threat, it is not clear that believing in these conspiratorial explanations alleviates the threat and makes people feel better [20, 21]. Indeed, van Prooijen noted that "[w]hile current theoretical models may imply that conspiracy theories satisfy basic psychological needs or help alleviate threats, empirical evidence does not support that people actually benefit from conspiracy beliefs in this manner" [22, p. 1].

In fact, the available research suggests that although conspiracy beliefs may provide some entertainment and purpose for believers [22], they actually have detrimental effects on stress and *well-being*—the degree to which a person experiences positive versus negative emotions and thinks about their life in a favorable manner [23, 24]. This is true for both an event conspiracy belief as well as more general conspiratorial ideation. Several studies have found that people experience greater lack of control, powerlessness, uncertainty, and disillusionment after being exposed to a specific conspiracy theory [16,25]. Research conducted during the COVID-19 pandemic paints a similar picture: belief in COVID-19 conspiracy theories assessed during an initial wave of data collection prospectively predicted decreases in well-being [26]. Conspiracist ideation—the general tendency to engage in conspiratorial thinking—appears to be similarly associated with an increase in learned helplessness, greater stress, and poorer well-being, among a variety of other undesirable mental health outcomes [27,28].

## Current research

Does believing in a given conspiracy theory not have any functional value for the believer? The central thesis guiding the current research is that believing specific conspiracy theories *can* have benefits for the believer, although such benefits may diminish over time. An analysis of the anatomy of conspiracy theories suggests they possess features suited to make people feel better in the face of turmoil: conspiracy theories provide believers with systematic (i.e., non-random) explanations for events that divide the world into camps of "good" (e.g., the believer and like-minded people) and "evil" (the group blamed for the conspiracy; [29]). As such, the motivated reasoning about the turmoil that is associated with conspiracy beliefs has the potential to alleviate threats to epistemic, existential, and social motives by legitimizing a narrative that provides a sense of understanding, security, and integrity [1]. Moreover, endorsing a given event conspiracy theory may make people feel unique and important, as well as provide a sense of excitement and alignment with social groups [22,30,31], which may ultimately influence well-being.

We contend that prior research has failed to uncover beneficial effects of believing an event conspiracy theory partly because such studies have neglected to consider two forces that operate concurrently with the development of an event conspiracy belief: (a) experiencing turmoil—the negative events that promote the development of the belief in the first place—and (b) conspiracist ideation—the individual's general level of conspiracist ideation. As such, we propose that experiencing turmoil and conspiracist ideation act as "suppressor" variables—those whose absence in a statistical model obscures the relation between a focal predictor and outcome [32,33] or changes the sign of that relation [34]. We elected to focus on these variables *a priori* because they are the most salient situational and dispositional factors in the development of event conspiracy beliefs in current research and theory in the field [1,10].

We elected not to include additional potentially relevant variables like political ideology and demographic characteristics because they seemed less able to account for the key relation of interest—that between an event conspiracy belief and well-being. For example, some prior work suggests that greater political extremity predicts greater belief in conspiracy theories [35], whereas other work suggests no systematic relation beyond greater belief in ideologically-congruent conspiracy theories [36]. Moreover, although age, education, income, and race are consistently linked to belief in conspiracy theories, these associations are, at best, modest [37]. In support of this choice, exploratory analyses that introduce political ideology and demographic variables into the relevant statistical models presented below render the results virtually unchanged (see S1 Data).

The theoretical framework guiding this research postulates that believing in an event conspiracy theory directly causes an increase in well-being—at least momentarily—but both

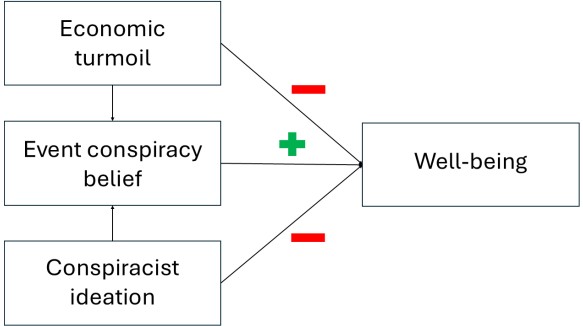

**Fig 1. Graphical representation of the proposed causal relation between event conspiracy beliefs, conspiracist ideation, the experience economic turmoil, and well-being.** Arrows represent proposed causal relations among variables (i.e., when considering unique influence of one variable on the other). The experience of economic turmoil and conspiracist ideation are proposed to exert an effect on both belief in an event conspiracy theory and well-being. Importantly, belief in an event conspiracy theory is proposed to exert its own unique effect on well-being. To be clear, this representation depicts the proposed psychological processes occurring at one point in time. Although there is evidence that belief in COVID-19 conspiracy theories can lead to increases in conspiracist ideation [44], the model limits itself to situations in which this recursive mechanism does not operate.

the experience of turmoil and conspiracist ideation exert causal effects on both well-being and belief in an event conspiracy theory, with the result being a suppression of the positive relation between event conspiracy beliefs and well-being (see Fig 1).

In support of the hypothesized beneficial effects of believing event conspiracy theories on well-being, we outline the evidence for other paths in the model below. First, there is evidence that both experiencing turmoil and conspiracist ideation promote belief in a given event conspiracy belief. Specifically, experiments have established a causal link between the experience of turmoil and distress to endorsement of more specific conspiracy theories [4,6]. Although experimental evidence is hard to come by given the practical and ethical concerns, conspiracist ideation robustly predicts more specific event conspiracy theories [9,10], which suggests that the general tendency to believe conspiracy theories leads people to be more likely to interpret explanations for a given event in a more conspiratorial manner. Moreover, both the experience of turmoil and conspiracist ideation are themselves also associated with greater stress and poorer well-being. Not surprisingly, experiencing economic turmoil is associated with decreases in well-being [38–41]. Similarly, conspiracist ideation has been linked to indicators of poorer well-being [26,28,42,43], although there are some conflicting findings in the literature [20]. Thus, the predicted positive relation between belief in event conspiracy theories and well-being may be obscured by the dual influence of turmoil and conspiracist ideation in increasing event conspiracy beliefs and lowering well-being, resulting in simple relations that are either null [21] or negative [27].

The need to account for the experience of turmoil when examining potential benefits of an event conspiracy belief seems relatively straightforward. However, readers may wonder why we propose that an event conspiracy belief can have benefits for people's level of stress and well-being, while the general tendency to believe conspiracy theories does not. We contend that, during a period of uncertainty and confusion like that created by the COVID-19 pandemic, individuals who endorse event-related conspiracy theories—logically flawed as they may sometimes be [45]—may fare better than those left without a clear causal explanation [46]. That is, non-conspiracy explanations, such as random chance or human error, mean the world is unpredictable, which is itself stressful [47]. When a group of people is conspiring, the world may seem less unpredictable. Moreover, such non-conspiracy explanations may feel

insufficient or trivial in the face of a global crisis due to proportionality bias—the tendency to believe that large effects stem from large causes [48]. Thus, a global shutdown may be more easily understood as being caused by the evildoings of global elite than by a virus spread in a wet market.

Conversely, higher levels of conspiracist ideation are unlikely to furnish such an overall benefit to the believer. The frequent tendency to see dark forces operating in numerous domains of one's life and the perseveration on those forces that accompany higher levels of conspiracist ideation is obviously stressful, as noted earlier [28]. Beyond a general focus on dark plots underlying a variety of events, conspiracist ideation may be unlikely to confer psychological benefits due to its association with learned helplessness and maladaptive coping strategies that are known correlates of poorer well-being [26,49].

Regarding the time course of our proposed effect, we predict that the benefits conferred by a specific event conspiracy belief are likely to dissipate over time. We reason that because believing conspiracy theories is linked to learned helplessness and maladaptive coping strategies [26,49] and has been connected to increases in conspiracist ideation over time [44], initial psychological benefits will be relatively short-lived. Nevertheless, we hypothesize that the adoption of a specific event conspiracy belief does convey some temporary functional value.

We have presented our conceptual reasoning in causal terms, highlighting the possibility that coming to believe a conspiracy theory about a specific event may benefit individuals' well-being. However, we wish to acknowledge from the very outset that any such causal inferences ultimately require experimental evidence. The studies we present are correlational in nature. Although we will report a number of analyses aimed at buttressing the proposed hypothesis, it should be kept in mind that the findings are merely consistent with our conceptual framework. Nevertheless, we hope that the findings will inspire further research that offers converging evidence for causal inferences.

In summary, we propose that believing conspiracy theories may indeed be associated with benefits for the believer, but that uncovering these associations requires disentangling the influence of endorsing specific conspiracy theories from the effects of (1) experiencing turmoil—enduring the distressing situation that nudges people toward conspiracy theories in the first place—and (2) conspiracist ideation—the more general tendency to believe conspiracy theories. Stated differently, we hypothesize that belief in an event conspiracy theory will predict beneficial outcomes when statistically accounting for people's experience of turmoil and their level of conspiracist ideation. Moreover, we hypothesize that the strength of the relation between belief in an event conspiracy theory and psychological benefits is likely to weaken over time. To test these ideas, we analyzed data from two research projects conducted during the COVID-19 pandemic [50, 51].

## Study 1

In Study 1, we used cross-sectional data to examine the relation between specific conspiracy beliefs and a variable known to be an important influence on well-being—people's general level of stress [52]. Specifically, we estimated the association between COVID-19 conspiracy beliefs and people's general level of stress, while accounting for their experience of turmoil and conspiracist ideation. To be specific, we assessed the experience of turmoil by measuring the degree to which a participant reported suffering economic consequences as a result of the COVID-19 pandemic. To assess conspiracist ideation, Study 1 relied on a widely-used and well-validated measure of the construct that taps into people's generic conspiracist beliefs—the Generic Conspiracist Beliefs Scale [9]. Belief in event conspiracy theories was assessed by asking people to what extent they endorsed two common conspiracy theories about

COVID-19—that COVID-19 was deliberately released and that the seriousness of COVID-19 was intentionally exaggerated [53]. As a proxy for well-being, we assessed people's reports regarding the general level of stress they were currently experiencing. Stress refers to an individual's appraisal of the demands placed on them in relation to their resources [54], and is related to but distinct from well-being [55].

## Methods

### Participants

Five hundred and one MTurk workers provided informed consent and completed a ten-minute survey in exchange for $1.00. The target sample size was selected to ensure stable estimates of correlation coefficients [51,56]. Participation was restricted to USA-based workers. To obtain high-quality data, only those workers with 500 + approved HITs and a minimum approval rate of 95% were eligible to participate in the study. As an additional layer of data quality, participants who failed an attention/comprehension check ($n = 60$) were excluded from the main analyses. Notably, the inclusion of participants who failed the attention/comprehension check in the analyses does not substantively alter the results. The final sample consisted of 441 participants (Gender: 201 female, 238 male, 1 gender identification not listed, 1 preferred not to answer; Age [years]: $M = 37.5$, $SD = 11.9$). A sensitivity analysis revealed that we were 80% powered to detect a small effect ($f^2 = 0.018$; [57]).

The data were collected using the Mechanical Turk (MTurk) platform on June 9, 2020. Given the manner through which the virus is transmitted and the infection rate in the United States of America at the time, online data collection was the only legal and ethical method of conducting the study. Moreover, using the MTurk platform did provide advantages over alternative methods of data collection, including greater demographic [58] and geographic diversity than is found in college student samples often used in psychological research. Importantly, MTurk samples generally perform similarly to samples drawn from other sources across many tasks [59]. Moreover, relations that have been reported elsewhere in the literature (e.g., between disgust sensitivity and perceived vulnerability to disease) successfully replicate in this sample [19,51].

### Materials

Among a variety of other instruments (see [51], for a full list), participants completed measures of economic turmoil stemming from the pandemic, conspiracist ideation, belief in COVID-19 conspiracy theories, and general stress. The Institutional Review Board at Ohio State University approved all study procedures. All relevant measures, manipulations, and exclusions are reported. Study materials, data, syntax, and codebook files can be retrieved online: https://osf.io/7b3xp/?view_only=f649bb67327c40629d9b63b3b2188fd5.

### Measures

#### Conspiracist ideation

Individual differences in the tendency to believe conspiracy theories were assessed using a shortened 10-item version of the Generic Conspiracist Beliefs Scale (GCBS; [9]). This trimmed version of the scale was employed to reduce participant burden, and was constructed by selecting the two items from each of the five factors of the scale with the highest item-total correlations [9]. Despite its shorter length, the modified GCBS employed in this study demonstrated excellent internal consistency in this study (α = 0.94) and validity in other research [51]. Participants rated each item of the scale (e.g., "Secret organizations communicate with

extra-terrestrials, but keep this fact from the public") on a five-point scale with the endpoints of *Definitely not true* and *Definitely true.*

The content of one item of the GCBS ("The spread of certain viruses and/or diseases is the result of the deliberate, concealed efforts of some organization") overlaps somewhat with the item assessing people's belief that the COVID-19 pandemic was human-made. Importantly, however, the results of the analyses presented in the main text remain substantively unchanged when this overlapping item is removed from the GCBS. See S1 Data for full details.

### Economic turmoil

The experience of economic turmoil due to the COVID-19 pandemic was measured using a one-item self-report of the degree to which they suffered negative economic consequences as a result of the pandemic ("To what degree have you and your immediate family personally suffered negative economic/financial consequences [e.g., laid-off, lost wages] because of the COVID-19/ coronavirus restrictions?"). This item was rated on a seven-point scale ranging from *1 = Not at all* to *7 = Very much.*

### Belief in COVID-19 conspiracy theories

Drawing inspiration from Imhoff and Lamberty's research on the relation between COVID-19 conspiracy theories and different patterns of pandemic-related behavior, we measured belief in COVID-19 conspiracy theories by asking participants to indicate how much they believed that (1) the COVID-19 pandemic was a hoax ("COVID-19 is intentionally presented as dangerous in order to mislead the public") and (2) COVID-19 was deliberately created and released ("COVID-19 was intentionally brought into the world for dark purposes") using a seven-point scale ranging from *-3 = Strongly disagree* to *+3 = Strongly agree* [53]. Because the two measures correlated strongly ($r = 0.80$), and given our interest in creating an index of the degree to which people believed conspiracy theories about the virus and pandemic, we averaged the two ratings to create a composite score. The results remain substantively unchanged when using each item assessing COVID-19 conspiracy beliefs as a predictor instead of the composite measure (see S1 Data for full details).

### General stress

As a proxy for people's well-being, we assessed the general level of stress they were experiencing at the moment via the item "Please rate how much stress you are currently experiencing in your daily life using the options below." This item was rated on a five-point scale anchored by the response options *No stress* and *An extreme amount of stress.*

## Results

Analyses were conducted using IBM SPSS Statistics 28 and R (v. 4.2.2) software. The inter-correlations between key variables are presented in Table 1. Before turning to the substantive analyses, we wanted to confirm that our essential contention that measures of specific conspiracy beliefs about COVID-19 and conspiracist ideation represent two related but distinct constructs. To this end, we first used parallel analysis [60] to determine the number of factors in these conspiracy item-related data. The results revealed two factors in the data. We followed this analysis by estimating a model using principal axis factoring with a direct oblimin (oblique) rotation, and requested the extraction of two factors. Consistent with our thinking, the results revealed that the conspiracist ideation items loaded onto the first factor, while the COVID-19 conspiracy items loaded onto a second factor. This second factor accounted for

**Table 1. Correlations among key variables in study 1.**

| | 1. | 2. | 3. | 4. |
|---|---|---|---|---|
| 1. Stress | — | | | |
| 2. COVID-19 Conspiracy Beliefs | -0.06 | — | | |
| 3. Experience of Economic Turmoil | 0.24** | 0.43** | — | |
| 4. Conspiracist Ideation | 0.02 | 0.71** | 0.42** | — |

Note. $N$ = 441.

**p <.01.

an additional 5.92% of the variance. As expected, the two factors were significantly correlated with each other, $r$ (439) = 0.66, $p$ < 0.001. Nevertheless, this analysis suggests that they are separable constructs (see S1 Data for full details).

## Multiple regression analyses

We turned next to the tests of our hypothesis. All variables were standardized (z-scored) prior to analysis. We used a series of hierarchical Ordinary Least Squares (OLS) regression models to examine the relation between belief in event conspiracy theories and stress. All variance inflation factor (VIF) values were well below commonly used thresholds for issues of multicollinearity (all VIFs < 2.4). In the initial step, we estimated the simple relation between endorsement of COVID-19 conspiracy theories and stress, which revealed no significant association, $\beta$ = -0.06 (95% CI: -0.16, 0.03), $t$(439) = -1.28, $p$ = 0.20. Nex$t$, we added reports of economic consequences due to the pandemic to the model. Unsurprisingly, experiencing greater economic consequences was associated with greater levels of stress, $\beta$ = 0.32 (95% CI: 0.22, 0.42), $t$(438) = 6.40, $p$ < 0.001. Impor$t$antly, when holding economic consequences constant, greater endorsement of COVID-19 conspiracy theories predicted *less* stress, $\beta$ = -0.20 (95% CI: -0.30, -0.10), $t$(438) = -3.99, $p$ < 0.001.

We followed this analysis by removing reports of economic consequences from the model and replacing it with people's conspiracist ideation scores. Here, we found a marginally-significant relation between conspiracist ideation and stress, such that greater conspiracist ideation was associated with greater stress, $\beta$ = 0.12 (95% CI: -0.008, 0.26), $t$(438) = 1.84, $p$ = 0.07, consistent with empirical literature summarized earlier regarding the negative consequences of conspiratorial thinking. However, when added to the equation, belief in COVID-19 conspiracy theories was, as in the preceding analysis, a significant predictor of less stress. That is, greater endorsement of COVID-19 conspiracy theories was associated with less stress, $\beta$ = -0.15 (95% CI: -0.28, -0.017), $t$(438) = -2.22, $p$ = 0.027.

Most importantly, in our final analysis, we reintroduced people's reports of economic consequences suffered due to COVID-19 into the model so that all three predictors were in the model simultaneously. This last analysis revealed that, after accounting for economic consequences and conspiracist ideation, greater belief in COVID-19 conspiracy theories predicted less stress, $\beta$ = -0.24 (95% CI: -0.37, -0.11), $t$(437) = -3.57, $p$ < 0.001. In summary, the critical relation between endorsement of COVID-19 conspiracy theories and stress changed from non-significance ($\beta$ = -0.06, $p$ = 0.20) to a statistically substantial level ($\beta$ = -0.24, $p$ < 0.001) after controlling for the two precursors—the experience of turmoil and conspiracist ideation.

We conducted an additional analysis to ensure the robustness of our results. Although not a part of our theoretical framework, we included political ideology, age, race, income, and education as additional covariates in a multiple regression analysis including all prior predictors. Even when controlling for political ideology and demographics (age, race, income,

and education; see [37]) in addition to turmoil and conspiracist ideation, belief in COVID-19 conspiracy theories continued to predict lower levels of stress (see S1 Data for details).

### Propensity score weighting analyses

Given our interest in testing whether event conspiracy beliefs exert a causal effect on stress, we performed an exploratory analysis using the propensity score balancing technique with stress as the outcome, belief in COVID-19 conspiracy theories as a continuous treatment variable, and our key covariates—the experience of economic turmoil and conspiracist ideation—as confounders. Propensity score weighting attempts to mimic characteristics of a randomized control trial by balancing the distribution of potential confounders across values of the focal predictor [61]. Entropy balancing weights were generated via the *WeightIt* and *cobalt* packages in R (v. 4.2.2). This procedure was successful in achieving balance, as evidenced by correlations between the continuous treatment and covariates approximating 0. The final effective adjusted sample for this analysis was 185, a decrease from the 441 participants used in the multiple regression analyses above. Importantly, the weighted linear regression revealed that belief in COVID-19 conspiracy theories significantly predicted stress such that those who endorsed conspiracy theories more reported less stress, $\beta = -0.19$, $SE = 0.05$, $t(438) = -3.77$, $p < 0.001$. The results remain substantively unchanged when including political ideology, age, race, income, and education as covariates in the models (see S1 Data).

When looking across analyses, then, there is a consistent relation between belief in COVID-19 conspiracy theories and stress once economic turmoil and conspiracist ideation are taken into account.

## Discussion

The results of Study 1 provide initial support for our hypothesis: when people's level of experienced turmoil and conspiracist ideation are taken into account, greater endorsement of specific conspiracy theories predicts less stress. That is, for two people who experienced the same level of economic turmoil and have the same level of conspiracist ideation, an increase in belief in COVID-19 conspiracy theories is associated with a decrease in stress. This suggests that believing conspiracy theories about a specific event can indeed be associated with psychological benefits for the believer, but uncovering this relation requires that researchers statistically separate the influence of experiencing turmoil and a general tendency to believe conspiracy theories from the belief in a conspiracy theory regarding the specific event. To our knowledge, this is the first instance in which a positive relation between believing conspiracy theories and stress has been documented.

Importantly, the evidence linking belief in an event conspiracy theory to reduced stress is strong from a statistical standpoint. Not only do the results reveal the predicted effect in a relatively large sample, the statistical evidence for the effect ($p < 0.001$) is impressive relative to even the most conservative standards [62]. However, a key limitation of Study 1 is the cross-sectional nature of the data, which seriously limits the ability to draw inferences about prospective relations among the variables. We address this limitation in Study 2.

## Study 2

In Study 2, we used data made publicly available by the COVID-19 Psychological Research Consortium (C19PRC; [50]) to provide another test of our hypothesis that believing specific conspiracy theories can benefit people's well-being. The C19PRC consists of a multi-disciplinary team of psychologists interested in understanding the short- and long-term impacts of the pandemic on people's mental health, as well as their attitudes and beliefs about

a variety of topics. To this end, the C19PRC collected a large, nationally representative sample from the United Kingdom during the early stages of the pandemic, and followed up with these individuals via multiple waves over an extended period of time.

Study 2 represents a more robust test of our reasoning. First, we can use our measure of belief in COVID-19 conspiracy beliefs to predict well-being at later timepoints. Second, these data extend Study 1 by allowing us to assess the relation between an event conspiracy belief and the broader construct of well-being much more closely. Because the dataset included a variety of measures related to well-being, Study 2 sheds light on how an event conspiracy belief may make people feel better—through boosting positivity, reducing negativity, and/or improving life satisfaction [23, 24].

The C19PRC dataset assessed the conceptual variables that were of interest in Study 1 using similar measures. Specifically, conspiracist ideation was measured in Wave 1 via the Conspiracy Mentality Questionnaire [13]—another widely-used and well-validated measure of the construct. Approximately one month later, Wave 2 of data collection included a one-item measure of the degree to which people were worried about how COVID-19 impacted their finances and a two-item measure of belief in COVID-19 conspiracy theories. Another wave of data collection, some 2.5 months later, allowed us to assess subjective well-being via a number of different measures including happiness, hopefulness, life satisfaction, and symptoms of depression and anxiety. Roughly one year after the first wave of data collection, yet another wave of data collection provided measures of subjective well-being once again, including happiness and hopefulness.

## Methods

### Participants

The C19PRC recruited English-speaking adults residing in the United Kingdom for participation in the project through Qualtrics [63]. The initial wave of data collection consisted of 2,025 participants. For the current analyses, we focus on the variables directly relevant for testing our hypothesis, which were collected during Waves 1 (March 23-28, 2020), 2 (April 22 – May 1, 2020), 3 (July 9-23, 2020), and 5 (March 24 – April 20, 2021) of the project. The available sample for the key analyses presented below consisted of 712 participants (Gender: 309 female, 401 male, 1 preferred not to say; Age [years]: $M$ = 52.79, $SD$ = 13.81). A sensitivity analysis revealed that the study was 80% powered to detect a small effect, $f^2$ = 0.011 [57].

According to the original researchers, attrition in this multi-wave study was predominantly predicted by demographic rather than psychological variables (see [64], for details).

### Materials

To access the publicly-available data and the accompanying study materials for the C19PRC study, please visit https://osf.io/v2zur/. For syntax specific to the analyses reported here, see: https://osf.io/7b3xp/?view_only=f649bb67327c40629d9b63b3b2188fd5.

### Measures

#### Conspiracist ideation—wave 1

Conspiracist ideation was assessed via the Conspiracy Mentality Questionnaire [13]. This instrument consists of five generic statements espousing a conspiratorial view of the world (e.g., "There are secret organizations that greatly influence political decisions"). Each item is rated using an 11-point scale anchored by the endpoints *Certainly not – 0%* and *Certainly 100%*. The scale showed good internal consistency (α = 0.85).

## Economic turmoil—wave 2

Paralleling Study 1, people's experience of economic turmoil was assessed by asking participants to reflect on their current level of worry about the manner in which their finances have been impacted by the pandemic ("On balance, how much are you worried about the way that your household finances have been affected by the coronavirus COVID-19 pandemic SO FAR?"). This single-item measure was rated on a 10-point scale ranging from *1. Not at all worried* to *10. Extremely worried.*

## Belief in empirically unsubstantiated claims/COVID-19 conspiracy theories—wave 2

The publicly available dataset upon which we relied did not include measures that directly assessed COVID-19 conspiracy beliefs. However, empirically unsubstantiated claims that underpin such conspiracy theories were assessed and, hence, served as our proxies for COVID-19 conspiracy beliefs. Specifically, we used two self-report items focused on people's endorsement of the ideas that (1) the severity of the COVID-19 pandemic was exaggerated ("Coronavirus is actually no more dangerous than the common flu.") and (2) COVID-19 was deliberately created ("Covid-19 was developed in a lab in Wuhan, China."). Both items were rated using a slider scale ranging from *Do not believe at all 0%* to *Completely believe 100%*. These two items were selected for use in Study 2 due to their similarity to those used to measure belief in COVID-19 conspiracy theories in Study 1. Although the two items composing this scale are more modestly correlated than in Study 1 ($r = 0.25$), the results remain substantively unchanged when using each individual item as a predictor instead of the composite measure (see S1 Data for full details).

  We acknowledge that the items above—particularly those involving the severity of COVID-19—more clearly assess endorsement of empirically unsubstantiated claims that underpin COVID conspiracy beliefs than conspiracy theories per se. That said, we wished to test whether these unsubstantiated claims could be interpreted as being indicative of conspiracy theories. To this end, we recruited an independent sample of 497 participants to provide ratings of the extent to which the items were indicative of believing conspiracy theories. First, we defined the term conspiracy theory in the manner done in this manuscript, highlighting that "a conspiracy theory involves an assertion that dark forces are at work." Next, participants were asked to consider whether believing various statements "would imply belief in a COVID conspiracy theory back during the pandemic." They then rated the degree to which a person who endorsed each of the two items above would believe a conspiracy theory regarding COVID-19. We also asked participants to rate two items intended to serve as comparison points, one clearly expressing a conspiracy theory ("Lockdowns were put in place during the COVID-19 pandemic deliberately for the purpose of hurting the economy") and one expressing a statement well-supported by evidence ("Social distancing is an effective method of preventing the spread of COVID-19"; [65]). The four items above were rated using a seven-point scale ranging from *Not at all* (1) to *Very much so* (7), with *Moderately so* (4) serving as the midpoint. Participants believed that during the height of the pandemic, people were more than moderately likely to believe a COVID-19 conspiracy theory when they endorsed the ideas that COVID-19 was no more dangerous than the flu ($M = 4.55$, $SD = 1.92$), $t(496) = 6.28$, $p < 0.001$, $d = 0.28$, and that COVID-19 was developed in a lab in Wuhan ($M = 5.34$, $SD = 1.88$), $t(496) = 15.92$, $p < 0.001$, $d = 0.71$. As points of reference, participants also believed that (a) those who endorsed the item regarding lockdowns having been intended to hurt the economy were more than moderately likely to endorse a conspiracy theory regarding COVID-19 ($M = 5.01$, $SD = 2.19$), whereas (b) those who endorsed the evidence-supported item regarding the

efficacy of social distancing were judged less likely to believe a COVID conspiracy theory ($M$ = 2.94, $SD$ = 2.08).

### Subjective well-being—wave 3

Given the multi-faceted nature of subjective well-being [23], a number of instruments were used to capture it. These included multi-item scales of happiness and hopefulness, a one-item measure of life satisfaction, and scales of depression and anxiety symptomology.

Happiness was assessed using the Subjective Happiness Scale [66], which consists of four items asking people to rank their general and relative level of happiness on a variety of seven-point scales (e.g., "In general, I consider myself to be:" *1 = Not a very happy person* to *7 = A very happy person*). The scale showed good internal consistency (α = 0.84).

The positively-worded Brief Measure of Hopelessness Scale (Brief-H-Pos) was used to measure hopefulness [67]. This two-item scale asks participants to rate the extent to which they feel hopeful about the future and their ability to accomplish their goals (e.g., "The future seems to me to be hopeful and I believe that things are changing for the better") using a five-point scale ranging from *1 = Absolutely disagree* to *5 = Absolutely agree.* This scale showed good internal consistency (α = 0.83).

To measure depression symptomology, participants completed the Patient Health Questionnaire nine-item depression scale (PHQ-9; [68]). On this instrument, people report how frequently they have experienced symptoms of depression (e.g., "Little interest or pleasure in doing things") over the past two weeks on a four-item scale ranging from *0 = Not at all* to *3 = Nearly every day.* The PHQ-9 showed excellent internal consistency (α = 0.93).

The seven-item Generalized Anxiety Disorder Scale (GAD-7; [68]) was employed to assess people's level of anxiety. In a manner parallel to that of the PHQ-9, people report on the frequency of anxiety symptomology over the past two weeks using a four-point scale ranging from *0 = Not at all* to *3 = Nearly every day.* The GAD-7 showed excellent internal consistency (α = 0.95).

Participants rated their current satisfaction with life ("Thinking about your life as it is right now, how satisfied are you with your life?") using a 100-point slider scale bookended by the markers *0 – Completely unsatisfied* and *100 – Completely satisfied.*

### Subjective well-being—wave 5

Once again, a number of scales were used to measure well-being. The same multi-item scales were used to tap into hopefulness, depression, and anxiety. In contrast to the prior wave of data collection, however, this measurement occasion employed a one-item measure of happiness and included no measure of life satisfaction.

Happiness was measured using a one-item scale asking people to report how happy they felt yesterday ("Overall, how happy did you feel yesterday, where 0 is 'not at all happy' and 10 is 'completely happy'").

As in Wave 3, the positively-worded Brief Measure of Hopelessness Scale (Brief-H-Pos) was used to measure hopefulness [67]. This scale showed good internal consistency (α = 0.88).

As in Wave 3, depression symptomology was measured using the PHQ-9 [68]. The PHQ-9 again showed excellent internal consistency (α = 0.92).

As in Wave 3, anxiety was assessed using the GAD-7 [68]. Once again, the GAD-7 showed excellent internal consistency (α = 0.95).

### Results

Analyses were conducted using IBM SPSS Statistics 28 and R (v. 4.2.2) software. To test our hypothesis regarding the unique positive impact of belief in COVID-19 conspiracy theories

on well-being, we pursued an analytic approach that relied on identifying latent factors among the dependent measures, and then regressing the resulting factor scores on our predictors of interest. Before conducting the exploratory factor analysis, we assessed the adequacy of our Wave 3 and Wave 5 data for the technique. For the Wave 3 data, the Kaiser-Meyer-Olkin measure of sampling adequacy was 0.95 and Bartlett's test of sphericity was significant, $\chi^2(253)$ = 13,689.17, $p < .0001$, both of which indicated that our data was indeed suitable for factor analysis. For the Wave 5 data, both the Kaiser-Meyer-Olkin measure of sampling adequacy (0.96) and Bartlett's test of sphericity, $\chi^2(171)$ = 11,783.87, $p < .001$, once again indicated that our data was suitable for factor analysis.

Affective positivity and negativity are the two core components of well-being [23, 24]. Accordingly, there was strong theoretical backing for the idea that our constituent outcome measures would load onto two factors. To start, we entered each individual item from the five measures of subjective well-being from Wave 3 into an exploratory factor analysis with a direct oblimin (oblique) rotation, which most clearly yielded a two-factor solution according to both eigenvalues and an inspection of the scree plot. Given the convergence between theory and the results of the factor analysis, we extracted two factors. The two factors extracted showed a negative correlation, $r(710)$ = -0.53, $p < 0.001$. As seen in Table 2, the items dealing with depression and anxiety symptomology loaded strongly on the first factor, which accounted for 48.6% of the variance. We labelled this the negative symptomology factor. The second factor accounted for an additional 10.0% of the variance, and was comprised of items assessing happiness, hopefulness, and life satisfaction, except for one item of the happiness scale, which did not load on either factor. As such, we labelled this the contentment factor. The next closest factor accounted for 3.9% of variance.

We performed a similar procedure to determine the factor structure of the well-being measures from Wave 5. All individual items from the four measures of subjective well-being from Wave 5 were entered into a factor analysis with a direct oblimin (oblique) rotation in which we requested the extraction of two factors. Once again, the two factors were negatively correlated, $r(710)$ = -0.53, $p < 0.001$. As expected, the factor structure for the Wave 5 data very much resembled that which was observed for Wave 3 (see S1 Data for details). Specifically, the first factor consisted of items dealing with depression and anxiety symptomology, which accounted for 56.4% of the variance. The second factor accounted for an additional 6.5% of the variance, and was comprised of items assessing happiness and hopefulness. In keeping with Wave 3 data, we labelled the first factor negative symptomology and the second factor contentment.

The intercorrelations between key variables are presented in Table 3.

## Multiple regression analyses

Next, we conducted a series of linear regressions that focused on predicting the scores derived from the factor analysis. All variables were standardized (z-scored) prior to analysis. All VIF values were well below commonly used thresholds for issues of multicollinearity (all VIFs < 1.2). Paralleling the analytic strategy employed in Study 1, we first estimated the simple relation between belief in COVID-19 conspiracy theories and the factor scores before adding economic turmoil to the model. Then, we replaced economic turmoil with conspiracist ideation in the model, and lastly, we included all three predictors (belief in COVID-19 conspiracy theories, economic turmoil, and conspiracist ideation) in the model.

We began by predicting negative symptomology in Wave 3. An initial model revealed that greater belief in COVID-19 conspiracy theories predicted greater subsequent negative symptomology, $\beta$ = 0.09 (95% CI: 0.02, 0.17), $t(710)$ = 2.50, $p$ = 0.01. When the experience

**Table 2. Results from a factor analysis of measure of subjective well-being in wave 3.**

| Item | Factor | |
|---|---|---|
| | **Negative symptomology** | **Contentment** |
| Being so restless that it is hard to sit still | .85 | |
| Trouble relaxing | .84 | |
| Not being able to stop or control worrying | .83 | |
| Feeling nervous, anxious or on edge | .82 | |
| Worrying too much about different things | .83 | |
| Trouble concentrating on things, such as reading the newspaper.. | .82 | |
| Feeling afraid as if something awful might happen | .80 | |
| Feeling down, depressed, or hopeless | .78 | |
| Feeling bad about yourself - or that you are a failure or have let… | .74 | |
| Moving or speaking so slowly that other people have noticed? Or the opposite.. | .74 | |
| Becoming easily annoyed or irritable | .73 | |
| Little interest or pleasure in doing things | .70 | |
| Feeling tired or having little energy | .69 | |
| Poor appetite or overeating | .69 | |
| Trouble falling or staying asleep, or sleeping too much | .66 | |
| Thoughts that you would be better off dead or of hurting yourself… | .64 | |
| In general, I consider myself to be: Not a very happy person to A very happy person | | .89 |
| Some people are generally very happy. They enjoy life regardless of what is going on, getting the most out of everything… | | .88 |
| Compared with most of my peers, I consider myself to be: *Less happy* to *More happy* | | .84 |
| I feel that it is possible to reach the goals I would like to strive for | | .69 |
| Thinking about your life as it is right now, how satisfied are you with your life? | | .69 |
| The future seems to me to be hopeful and I believe that things are changing… | | .65 |
| Some people are generally not very happy. Although they are not depressed, they never seem as happy… | | |

**Table 3. Correlations among key variables in Study 2.**

| | 1. | 2. | 3. | 4. | 5. | 6. | 7. |
|---|---|---|---|---|---|---|---|
| 1. Negative symptomology - Wave 3 | — | | | | | | |
| 2. Contentment - Wave 3 | -.48** | — | | | | | |
| 3. Negative symptomology - Wave 5 | .77** | -.48** | — | | | | |
| 4. Contentment - Wave 5 | -.41** | .67** | -.51** | — | | | |
| 5. COVID-19 Conspiracy Beliefs | .10** | .11** | .09* | .01 | — | | |
| 6. Economic Turmoil | .24** | -.12** | .25** | -.18** | .18** | — | |
| 7. Conspiracist Ideation | .14** | -.01 | .13** | -.05 | .24** | .13** | — |

Note. $N = 712$. ** $p < .01$. * $p < .05$.

of economic turmoil was added to the model, greater economic distress predicted greater negative symptomology, $\beta$ = 0.23 (95% CI: 0.16, 0.30), $t(709)$ = 6.16, $p$ < 0.001. In this model, the relation between belief in COVID-19 conspiracy theories and negative symptomology was rendered non-significant, $\beta$ = 0.05 (95% CI: -0.02, 0.13), $t(709)$ = 1.45, $p$ = 0.15. We then replaced economic turmoil with conspiracist ideation in the regression model. As expected, higher levels of conspiracist ideation were associated with greater negative symptomology, $\beta$ = 0.12 (95% CI: 0.04, 0.19), $t(709)$ = 3.00, $p$ = 0.003. Moreover, the relation between endorsement of COVID-19 conspiracy theories and negative symptomology was reduced in magnitude relative to the base model, becoming marginally-significant, $\beta$ = 0.07 (95% CI: -0.009, 0.14), $t(709)$ = 1.72, $p$ = 0.09. Most importantly, however, when accounting for both conspiracist ideation and economic turmoil, belief in COVID-19 conspiracy theories was no longer significantly associated with subsequent negative symptomology, $\beta$ = 0.03 (95% CI: -0.04, 0.11), $t(708)$ = 0.85, $p$ = 0.39. In sum, the critical relation between endorsement of COVID-19 conspiracy theories and negative symptomology changed from a statistically-significant level ($\beta$ = 0.09, $p$ = 0.01) to non-significance ($\beta$ = 0.03, $p$ = 0.39), after controlling for the two precursors.

We repeated the same analytic strategy when predicting Wave 3 scores for the contentment factor. We found that greater belief in COVID-19 conspiracy theories predicted greater subsequent contentment, $\beta$ = 0.10 (95% CI: 0.03, 0.18), $t(710)$ = 2.73, $p$ = 0.006. With economic turmoil introduced into the model along with belief in COVID-19 conspiracy theories, greater turmoil predicted less subsequent contentment, $\beta$ = -0.14 (95% CI: -0.21, -0.06), $t(709)$ = -3.67, $p$ < 0.001. At the same time, belief in COVID-19 conspiracy theories continued to significantly predict subsequent contentment, $\beta$ = 0.13 (95% CI: 0.05, 0.20), $t(709)$ = 3.35, $p$ < 0.001.

In a subsequent step, economic turmoil was replaced by conspiracist ideation in the model. Descriptively, greater conspiracist ideation was associated with less contentment, $\beta$ = -0.06 (95% CI: -0.13, 0.02), $t(709)$ = -1.44, $p$ = 0.15—although this relation did not reach statistical significance. However, belief in COVID-19 conspiracy theories remained a significant predictor of subsequent contentment, $\beta$ = 0.12 (95% CI: 0.04, 0.19), $t(709)$ = 3.00, $p$ = 0.003. Lastly, when all predictors were included in the model, greater belief in COVID-19 conspiracy theories predicted greater future contentment, $\beta$ = 0.14 (95% CI: 0.06, 0.21), $t(708)$ = 3.52, $p$ < 0.001. In other words, a positive relation between endorsement of COVID-19 conspiracy theories was evident initially ($\beta$ = 0.10, $p$ = 0.006) and only grew stronger after taking conspiracist ideation and economic distress into account ($\beta$ = 0.14, $p$ < 0.001). The results remain substantively unchanged when using each item assessing COVID-19 conspiracy beliefs as a predictor instead of the composite measure (see S1 Data for full details).

Finally, we turn to the analysis of outcomes derived from Wave 5, which took place nearly one year after the pandemic's start. Following the same analytic strategy employed before, we found that when entered in the model alone, belief in COVID-19 conspiracy theories positively predicted negative symptomology, $\beta$ = 0.08 (95% CI: 0.01, 0.15), $t(710)$ = 2.23, $p$ = 0.03. Adding negative economic consequences to the model revealed that suffering economic turmoil at the start of the pandemic significantly predicted greater negative symptomology one year later, $\beta$ = 0.25 (95% CI: 0.17, 0.32), $t(709)$ = 6.69, $p$ < 0.001. Interestingly, when negative economic consequences are added to the model, belief in COVID-19 conspiracy theories is no longer associated with negative symptomology, $\beta$ = 0.04 (95% CI: -0.03, 0.11), $t(709)$ = 1.09, $p$ 0.28.

Replacing the experience of negative economic consequences in the regression model with conspiracist ideation reveals that a stronger general tendency to believe conspiracy theories significantly predicts greater negative symptomology later on, $\beta$ = 0.11 (95% CI: 0.04, 0.19),

$t(709) = 2.95$, $p = 0.003$, but COVID-19 conspiracy beliefs are unrelated to negative symptomology, $\beta = 0.06$ (95% CI: -0.02, 0.13), $t(709) = 1.46$, $p = 0.15$. When all three predictors were entered into the model simultaneously, belief in COVID-19 conspiracy theories did not significantly predict negative symptomology, $\beta = 0.02$ (95% CI: -0.06, 0.09), $t(708) = 0.52$, $p = 0.61$. Mirroring the findings of Wave 3, the relation between COVID-19 conspiracy beliefs and negative symptomology transformed from being highly statistically significant ($\beta = 0.25$, $p < 0.001$) to non-significance ($\beta = 0.02$, $p = 0.61$), when controlling for the two precursors.

Lastly, we turned our attention to the contentment factor from Wave 5 of data collection. In contrast to the analysis reported for Wave 3 data, there was no simple relation between belief in COVID-19 conspiracy theories and contentment at Wave 5, $\beta = -0.01$ (95% CI: -0.08, 0.06), $t(710) = -0.27$, $p = 0.79$. Adding negative economic consequences due to the pandemic to the mode revealed that economic turmoil at the onset of the pandemic significantly predicted less contentment a year later, $\beta = -0.19$ (95% CI: -0.26, -0.12), $t(709) = -5.02$, $p < 0.001$, but COVID-19 conspiracy beliefs remained unrelated to contentment, $\beta = 0.02$ (95% CI: -0.05, 0.09), $t(709) = 0.61$, $p = 0.54$.

Replacing negative economic consequences with conspiracist ideation in the model revealed neither conspiracist ideation, $\beta = -0.06$ (95% CI: -0.14, 0.01), $t(709) = -1.63$, $p = 0.10$, nor COVID-19 conspiracy beliefs significantly predicted contentment, $\beta = 0.005$ (95% CI: -0.07, 0.08), $t(709) = 0.13$, $p = 0.90$. When all three predictor variables were in the model, belief in COVID-19 conspiracy theories did not predict contentment, $\beta = 0.03$ (95% CI: -0.04, 0.11), $t(708) = 0.86$, $p = 0.39$.

As in Study 1, we checked the robustness of our results by including political ideology and demographic variables (age, race, income, and education) into the regression models discussed above. The inclusion of these additional covariates resulted in no changes to the substantive interpretation of the results. In particular, belief in COVID-19 conspiracy theories continued to predict greater contentment at Wave 3 even when statistically accounting for the experience of turmoil, conspiracist ideation, political ideology, and demographic characteristics.

## Mixed model comparing regression coefficients across time

Comparing this pattern of Wave 5 relations between COVID-19 conspiracy beliefs and contentment reveals a stark contrast to those found in Wave 3. Whereas the Wave 3 data revealed a statistically significant relation between belief in COVID-19 conspiracy theories and contentment ($\beta = 0.10$, $p = 0.006$) that only became stronger after accounting for the other precursors ($\beta = 0.14$, $p < 0.001$) the Wave 5 data revealed a non-significant relation, both when COVID-19 conspiracy beliefs were entered alone in the model ($\beta = -0.01$, $p = 0.79$) and when entered along with conspiracist ideation and negative economic consequences ($\beta = 0.03$, $p = 0.39$). A repeated measures linear model revealed that when accounting for conspiracist ideation and negative economic consequences, the relation between belief in COVID-19 conspiracy theories and contentment at Wave 3 was significantly stronger than that at Wave 5, $F(1, 708) = 10.35$, $p = 0.001$, $\eta^2 = 0.01$ (all other $p$'s > 0.14). This latter finding indicates that the benefits with respect to contentment that were evident at Wave 3 had dissipated substantially nine months later.

## Propensity score weighting analyses

Once again, we used propensity score weighting analyses to further test our hypothesis regarding the beneficial effect of believing COVID-19 conspiracy theories on well-being. Specifically, we focused on using the propensity score balancing technique with contentment at Wave 3 as the outcome, belief in COVID-19 conspiracy theories as a continuous treatment variable, and our key covariates—the experience of economic turmoil and conspiracist ideation—as

confounders. As in Study 1, entropy balancing weights were generated via the *WeightIt* and *cobalt* packages in R (v. 4.2.2), which successfully achieved balanced, as indicated by correlations between the continuous treatment and covariates approximating 0. The final sample for this analysis was 659, a decrease from the initial 712 participants present in the analyses involving multiple linear regression. Paralleling those multiple regression results, the weighted linear regression revealed that belief in COVID-19 conspiracy theories significantly predicted greater contentment at Wave 3, $\beta$ = 0.15, $SE$ = 0.04, $t(710)$ = 3.93, $p$ < 0.001. The results remain substantively unchanged when including political ideology, age, race, income, and education as covariates in the models (see S1 Data).

Looking across analyses, a clear picture emerges: belief in COVID-19 conspiracy theories predicts greater contentment in the short-term, but this effect dissipates over time. No relation emerges between belief in COVID-19 conspiracy theories and negative symptomology. Importantly, these findings hold when using both multiple linear regression and propensity score weighting and when using only economic turmoil and conspiracist ideation as covariates or using these covariates in addition to political ideology and demographic characteristics.

## Discussion

In Study 2, we extended our prior findings by examining how belief in COVID-19 conspiracy theories relates to different aspects of well-being and how these relations change over time. Interestingly, the initial steps of the regression models involving Wave 3 data revealed that greater belief in COVID-19 conspiracy theories was significantly associated with more negative symptomology but also more contentment. However, when we examined the *unique* role of belief in an event conspiracy theory by accounting for the experience of economic turmoil and people's conspiracist ideation through a series of regression models, we found that belief in COVID-19 conspiracy theories consistently predicted greater contentment. One way to interpret this finding is that for two people who experienced the same level of economic turmoil and have the same level of conspiracist ideation, an increase in belief in COVID-19 conspiracy theories is associated with an increase in contentment.

Moreover, this regression model revealed that the relation with negative symptomology was weakened and no longer significant when considering economic turmoil and conspiracist ideation. These results suggest that the link established between event conspiracy beliefs and increased negative symptomology may actually be driven by the covariation between event conspiracy beliefs and the forces that operate at the same time as the development of an event conspiracy belief: (a) the turmoil that promotes the belief and (b) the individual's general tendency to believe conspiracy theories. Most importantly, this study provides evidence that believing in specific conspiracy theories *can* be associated with boosts in well-being (but see the limitations subsection in the Discussion section below for the assumptions required for this interpretation) and clarifies that such relations are specific to increased contentment for the believer.

The data from Wave 5 allowed to us examine how these relations changed over time. With regard to negative symptomology, the results largely mirrored those of Wave 3: greater belief in COVID-19 conspiracy theories predicted greater negative symptomology when considered alone, but this relation was rendered non-significant when taking into account conspiracist ideation and the experience of negative economic consequences. The pattern of relations between COVID-19 conspiracy beliefs and contentment obtained in Wave 5, however, sharply contrasted with those of Wave 3. In Wave 5, belief in COVID-19 conspiracy theories no longer predicted contentment, either when considered alone or when in the context of the other

precursors. It appears that the benefits to contentment conferred by believing specific conspiracy theories are short-lived. We reason that because believing specific conspiracy theories is associated with maladaptive coping responses [49] and increases in conspiracist ideation over time [44], any positivity that specific conspiracy beliefs bring to someone's life eventually dissipates, and only maladaptive coping and belief systems remain.

Rounding out the picture, Study 2 also revealed that experiencing economic turmoil due to the pandemic prospectively predicted poorer well-being in the form of both greater negative symptomology and less contentment, both a few months after the start of the pandemic and one year later. This sensible finding establishes that the psychological distress experienced early in the pandemic was sufficient to influence well-being months later. By contrast, conspiracist ideation—the general tendency to believe conspiracy theories—significantly predicted only subsequent negative symptomology, and not contentment, at both timepoints.

## General discussion

Across two studies, one cross-sectional in nature and the other prospective, we find converging evidence that believing specific conspiracy theories is associated with greater well-being. Specifically, Study 1 suggests that when people's experience of economic turmoil due to the pandemic and their level of conspiracist ideation are taken into account, belief in COVID-19 conspiracy theories predicts less concurrent general stress. Study 2 extends this finding by demonstrating that, after controlling for economic distress and conspiracist ideation, greater belief in COVID-19 conspiracy theories no longer prospectively predicts increases in negative symptomology and, more importantly, is all the more strongly predictive of subsequent increases in contentment. However, this relation between specific conspiracy beliefs and contentment lessens over time. Putting these findings together suggests that any association between believing specific conspiracy theories and well-being is restricted to positive affect—which may make people feel more capable of dealing with the threats they face—but only for a limited amount of time. To our knowledge, this is the first research to demonstrate that a positive relation between an event conspiracy belief and well-being.

We interpret the results as providing robust evidence for our hypothesis of a relation between event conspiracy beliefs and well-being. However, the nature of the results in Study 2—in which COVID-19 conspiracy beliefs selectively predict contentment in the short-term but not the long-term—may also be interpreted as indicating that the relation between event conspiracy beliefs and well-being is at least somewhat tentative. As with all research documenting novel relations, independent replication is required.

The current work provides an alternative perspective on the prior research that has tackled this question. An examination of the simple relations between belief in COVID-19 conspiracy theories and outcomes of interest in our study mirrors the heterogeneity that is observed in the literature. Whereas prior work has sometimes found that an event conspiracy belief does not yield any benefits when considered on its own [21]—a finding paralleled in Study 1 with the null relation with stress—other work has found that an event conspiracy theory is associated with poorer outcomes [27]—as in Study 2 where believing COVID-19 conspiracy theories predicted greater negative symptomology in the initial regression model. Importantly, however, when these simple relations are compared to the results of the multiple regression in which economic distress and conspiracist ideation are included as covariates in the model, the picture changes and the importance of considering the influence of these covariates becomes clear: belief in COVID-19 conspiracy theories predicts *less* stress in Study 1 and no longer predicts negative symptomology in Study 2. Importantly, however, the time course of the

measurements also seems to matter, as one year after the start of the pandemic, the benefits of COVID-19 conspiracy beliefs were no longer detectable.

Although some readers may doubt that believing seemingly-wild conspiracy theories can bring positivity into one's life, considering the words of former conspiracy theorists suggests that our findings match their personal experience. For example, in an interview with *Politico,* a former QAnon believer described the initial and long-term effects of believing conspiracy theories in the following manner:

> Initially, believing in Q felt amazing, like being in some sort of mystical state or euphoria. For about six weeks, my fears about impending doom because of Covid-19, climate change and what I perceived as the threat of fascism were gone. The world felt safe and I felt energized, confident, creative and brimming with love. I'm not religious, but I kept thinking "Thank you, God. Thank you, God. Thank you, God." I heard "Amazing Grace" playing in my mind… My initial QAnon euphoria wore off after a few weeks, leaving me with unease about the dark world controlled by the Cabal. [69]

Before discussing the broader implications of this work, we wish to highlight three methodological aspects of this work we regard as important. First, given the lack of experimental evidence, strong causal claims are unwarranted. Establishing causality requires demonstrating evidence of covariation, temporal precedence, as well as ruling out plausible alternative explanations [70]. Across two studies, we provide evidence of covariation between event conspiracy beliefs and well-being. Although not definitive, the prospective nature of Study 2 provides some evidence regarding temporal precedence of the proposed relation between event conspiracy beliefs and well-being. Importantly, interpreting our results as unbiased estimates of the causal effect of event conspiracy beliefs on well-being relies on meeting a number of different assumptions. Naturally, a failure to rule out potential third variables would bias the estimates of the relation between event conspiracy beliefs and well-being, increasing the possibility of a spurious correlation. Although this research cannot definitively rule out all alternative explanations, it does leverage multiple regression and propensity score weighting analyses to rule out the most relevant situational, dispositional, and demographic variables as competing explanations based on the current state of the literature. In so doing, we furnish evidence that is consistent with the idea that believing specific conspiracy theories can have benefits for well-being. While it is possible that confounders other than those tested exist, we are unaware of likely candidates for the role. Moreover, the interpretation of results as unbiased estimates of a causal effect also relies on the degree of measurement error in confounding variables [71]. A failure to account for measurement error can lead to inflated Type I error rates [72]. One solution to this issue is the use of structural equation models (SEM), a method that estimates measurement error for multi-item measures. As a robustness check, we replicated our multiple regression analyses using SEM and found that our results remained substantively unchanged (see S1 Data). Although these results provide another converging line of evidence supporting our conclusions, our single-item measures of economic turmoil preclude us from modeling measurement error in this regard. Despite finding robust associations between economic turmoil and well-being, one valuable aspect of a future replication attempt would be the measurement of economic turmoil using a multi-item measure to ensure that this construct is assessed more comprehensively and measurement error can be modeled. Lastly, one additional assumption that enables an interpretation of our results as unbiased estimates of the causal effect involves is that the statistical models employed do not control for mediator or collider variables (see [71] for a full treatment of the assumptions that underlie interpreting observational data as an estimate of a causal effect). Inclusion of such variables in a model can

either obscure or change the association between predictor and criterion variables. We provide a comprehensive explanation of our rationale for the causal ordering of variables in the introduction, which suggests that the confounders in our models do not act as either mediators or colliders. Nevertheless, it should be kept in mind that the current findings represent merely a step, albeit an important one in our view, in building a case for a causal relation. The research presented here establishes a positive association between event conspiracy beliefs and well-being.

Second, skeptical readers may wonder about the interpretation of our measures of COVID-19 conspiracy beliefs in the context of our models, particularly with regard to conspiracist ideation. People higher in conspiracist ideation would be expected to endorse COVID-19 conspiracy theories to a greater extent by the nature of their general predisposition toward such explanations. Despite this predisposition, aspects related to the specific situation and how it impacts an individual contribute to the degree to which they endorse an event conspiracy theory. Indeed, individuals may experience situational pressures that spur the development of an event conspiracy belief [73–75] even those relatively low in conspiracist ideation [14]—which is how we view the postulated role of economic turmoil in the present work. Conceptually, we interpret belief in COVID-19 conspiracy theories when taking conspiracist ideation into account as representing people's belief that the events surrounding COVID-19 involved a conspiracy theory that are divorced from their general tendency to believe conspiracy theories and their associated constituents like anti-elitism and distrust of authorities. That is, we seek to isolate the aspect of endorsement of COVID-19 conspiracy theories that is unique to people's reactions to the COVID-19 pandemic rather than their general tendency to accept conspiratorial explanations. In effect, conspiracist ideation offers what might be considered a default level of acceptance of any given event conspiracy belief, but variability around that default is to be expected. For example, an individual whose conspiracist ideation focuses largely on distrust of government officials may accept a conspiratorial belief regarding a specific event involving the government readily, but may be less prone to accept a conspiratorial belief regarding a scientifically-supported medical practice. In the context of a regression equation including belief in COVID-19 conspiracy theories and conspiracist ideation, the model produces an index of belief in COVID-19 conspiracy theories that is divorced from an individual's general tendency to engage in conspiratorial thinking. This variance in COVID-19 conspiracy beliefs that is independent of conspiracist ideation is meaningful because the general tendency to accept conspiracy theories is not the only source of variance in a specific event conspiracy belief. More broadly, multiple regression allows us to obtain an estimate of the influence of COVID-19 conspiracy beliefs on well-being when holding the experience of economic turmoil and conspiracist ideation constant. In this way, it provides an index of the *unique* relation between COVID-19 conspiracy beliefs and well-being with the influence of economic turmoil and conspiracist ideation removed. To reiterate an earlier point, one way to interpret our findings is that for two people who have experienced the same amount of economic turmoil and have the same level of conspiracist ideation, an increase in endorsement of COVID-19 conspiracy theories during the COVID-19 pandemic was associated with an increase in well-being in the form of reduced stress and increased contentment.

Lastly, one of our key findings in Study 2 is a positive simple relation between belief in COVID-19 conspiracy theories and contentment that increases in magnitude once economic distress and conspiracist ideation are added to the model in Wave 3. This finding is particularly noteworthy because it centers on a positively-focused outcome, which stands in contrast to the vast majority of the prior work examining the costs and benefits of a *specific* event conspiracy belief. That is, one possible explanation for the lack of documentation of beneficial effects of believing specific conspiracy theories in prior research is that much of this work has

focused on negative symptomology as the key criterion (e.g., anxiety, uncertainty). Thus, this prior research and its focus on negativity may have missed an alternative pathway by which psychological benefits for the believer can be achieved: a boost in current contentment and hope regarding the future.

Although not central to the current research, we briefly note that some have suggested that there are important differences between commonly employed measures of conspiracist ideation [73]. Measures like the Generic Conspiracist Beliefs Scale [9] that was employed in Study 1 tap into the degree to which people endorse generic assumptions about the typicality of conspiratorial activity in the world (e.g., "Some UFO sightings and rumors are planned or staged in order to distract the public from real alien contact"). In contrast, instruments like the Conspiracy Mentality Questionnaire [13] that was employed in Study 2 tap into the broader ideological mindset of conspiracy mentality (sometimes referred to as a "generalized political attitude; [76]) by assessing the degree to which people endorse aspects of conspiracist thinking (e.g., "I think that there are secret organizations that greatly influence political decisions"). For simplicity, and given the substantial empirical relation between these two measures [10], we refer to what is measured by each approach as conspiracist ideation—people's general tendency to believe conspiracy theories. Moreover, the parallel findings that we observed with the two scales offers additional convergent validity for the conceptual framework.

Turning to this work's implications, there is a large body of research documenting the negative effects of believing conspiracy theories—both in the form of endorsement of specific conspiracy theories and the general tendency to believe conspiracy theories—for both the believer and those around them (e.g., enactment of risky health behaviors; [15,51]. We would like to stress that the current findings in no way invalidate the clear downsides that believing specific conspiracy theories have for the believer and those around them. Indeed, although our work suggests that believing a specific event conspiracy theory may make a person feel better in the moment, these effects disappear over time. Importantly, a growing body of research suggests that conspiracy beliefs have very real dangers for believers and those around them (see [8], for a review).

Looking more broadly, this research may also identify one important reason why people continually gravitate toward conspiracy theories—particularly in troubling times: they have the potential to make individuals feel better in the face of difficult circumstances. That is, our work highlights that developing a specific event conspiracy belief may serve as a means of coping with the precipitating threat, and hence, has some functional value for the believer. Acknowledging these potential benefits for believers of conspiracy theories not only provides a more complete picture of the psychology of conspiracy beliefs, but also is crucial for combating the spread of conspiracy theories. Our hope is that this work spurs interest among other researchers to extend our findings and better understand the nature of these effects.

## Supporting information

**S1 Data. A collection of supplemental analyses.**
(DOCX)

## Acknowledgements

We would like to thank Shelby T. Boggs, Jonathan L. Stahl, and Kaitlin M. Whitman for providing helpful feedback on an earlier version of this manuscript.

## Author contributions

**Conceptualization:** Javier A. Granados Samayoa.

**Formal analysis:** Javier A. Granados Samayoa.

**Funding acquisition:** Russell H. Fazio.

**Investigation:** Javier A. Granados Samayoa, Courtney A. Moore, Benjamin C. Ruisch, Jesse T. Ladanyi, Russell H. Fazio.

**Methodology:** Javier A. Granados Samayoa, Courtney A. Moore, Benjamin C. Ruisch, Jesse T. Ladanyi, Russell H. Fazio.

**Project administration:** Javier A. Granados Samayoa, Courtney A. Moore, Benjamin C. Ruisch, Jesse T. Ladanyi, Russell H. Fazio.

**Resources:** Javier A. Granados Samayoa, Courtney A. Moore, Benjamin C. Ruisch, Jesse T. Ladanyi, Russell H. Fazio.

**Supervision:** Russell H. Fazio.

**Validation:** Russell H. Fazio.

**Writing – original draft:** Javier A. Granados Samayoa.

**Writing – review & editing:** Javier A. Granados Samayoa, Courtney A. Moore, Benjamin C. Ruisch, Jesse T. Ladanyi, Russell H. Fazio.

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
