## [Decision Letter · Decision Letter 0]

23 May 2024

PONE-D-24-11501Is there anything good about conspiracy beliefs? Belief in COVID-19 conspiracy theories can provide benefits to well-beingPLOS ONE

Dear Dr. Fazio,

Thank you for submitting your manuscript to PLOS ONE. After careful consideration, we feel that it has merit but will need some revisions before a decision of acceptance. Therefore, we invite you to submit a revised version of the manuscript that addresses the points raised during the review process.

The comments from all three reviewers are quite constructive and could be addressed with thorough revisions. R1 praises the manuscript for its significant contribution to understanding the effects of conspiracy theory beliefs on well-being, suggesting minor revisions to strengthen the argument about the benefits of such beliefs. R2 and R3 both raise concerns about the manuscript's causal claims from correlational data, proposing a more rigorous justification for the causal model(s), clearer theoretical definitions, and some additional analyses to support the conclusions. 

We look forward to receiving your revised manuscript.

Kind regards,

Cengiz Erisen

Academic Editor

PLOS ONE

Additional Editor Comments:

Reviewers' comments:

Reviewer's Responses to Questions

**Comments to the Author**

1. Is the manuscript technically sound, and do the data support the conclusions?

Reviewer #1: Yes

Reviewer #2: Partly

Reviewer #3: Partly

2. Has the statistical analysis been performed appropriately and rigorously? 

Reviewer #1: Yes

Reviewer #2: Yes

Reviewer #3: I Don't Know

3. Have the authors made all data underlying the findings in their manuscript fully available?

Reviewer #1: Yes

Reviewer #2: Yes

Reviewer #3: Yes

4. Is the manuscript presented in an intelligible fashion and written in standard English?

Reviewer #1: Yes

Reviewer #2: Yes

Reviewer #3: Yes

5. Review Comments to the Author

Reviewer #1: This manuscript uses a cross-sectional survey and a multiple wave panel survey to test the hypothesis that belief in specific conspiracy theories (CTs) – specifically, belief in COVID-19 CTs – will have a positive impact on the believer’s well-being after controlling for the general tendency to believe CTs and the extent to which the person feels turmoil as a result of the event that gave rise to the specific CT belief in the first place. This is an easy review to write. Put succinctly, I think this is a terrific manuscript that makes an important contribution to the literature on the consequences of specific CT beliefs. I recommend its publication in PNAS.

I do have a few suggestions for minor revisions:

1. I think the authors’ argument regarding why belief in specific CTs can have benefits for individuals’ level of stress and well-being could be stronger. As it stands, there are only a few sentences about this (in the first paragraph under the “Current Research” heading and on the top of pg 10: “We contend that in the face of the kind of turmoil created by the COVID-19 pandemic…”) I recommend that the authors build this up a bit more, as this is the core of their argument – that belief in specific CTs can, in fact, be beneficial to individuals’ well-being (even if the benefits don’t last all that long).

2. I’m not sure it’s necessary to show the impact of specific CT beliefs while separately controlling for conspiratorial thinking and turmoil, then showing the effects while controlling for both conspiratorial thinking and turmoil in the same model. The authors aren’t interested in teasing apart the relative impact of the two variables (conspiratorial thinking or turmoil). As such, I think the authors can streamline the presentation of their results by focusing only on the models controlling for both variables. The other models could be moved to an appendix.

3. With regard to the relationship between conspiratorial thinking and (poorer) well-being, the authors could also site Farhart et al. (2021):

Farhart, Christina E., Joanne M. Miller, and Kyle L. Saunders. 2021. “Conspiracy Stress or Relief? Learned Helplessness and Conspiratorial Thinking.” In The Politics of Truth in Polarized America, ed. David Barker and Elizabeth Suhay, Oxford: Oxford University Press.

Reviewer #2: My thanks to the authors and editor for the opportunity to review this manuscript. It presents a cross-sectional study and a longitudinal study pertaining to the possible benefits of beliefs in specific COVID-19 conspiracy theories for wellbeing.

I found a lot to like about this manuscript. I love the premise: If people believe in conspiracy theories for specific motives, we should indeed be able to detect some benefits of beliefs in such theories (not just negative consequences). I thought the writing was admirably clear, and the analyses very thorough and comprehensively described. I also appreciated the sharing of open data and analysis scripts.

Despite these positive points, I have one core (and relatively substantial) concern about the manuscript, and it relates to the topic of causal inference. The manuscript is very clearly motivated by questions about the causal effects of belief in specific conspiracy theories on wellbeing. For example, the authors say, “The central thesis guiding the current research is that believing specific conspiracy theories can have benefits for the believer, although such benefits may diminish over time.” To produce a benefit is to have a causal effect. The abstract and discussion also make anumber of implicit causal inferences, e.g., “The current research provides evidence for benefits of endorsing specific conspiracy theories”; “conspiracy beliefs do provide at least temporary intrapersonal benefits”; “Across two studies, one cross-sectional in nature and the other prospective, we find converging evidence that believing specific conspiracy theories can have psychological benefits for the believer.” Yet the data is all purely observational/correlational. The authors acknowledge this by saying “given the correlational nature of this research, strong causal claims are unwarranted”, yet the manuscript does make strong causal claims (indeed, there’s one in the title). This presents a problem in terms of the relationship between the evidence presented and the claims made.

Obviously Study 2 is longitudinal, but longitudinal data isn’t a magic bullet for causal inference; its capacity to support causal claims depends on the nature of the data and the analyses conducted. E.g., some specific longitudinal designs and analyses (e.g., interrupted time series designs, random intercept cross lagged panel models) can permit at least some types of alternative explanations to be ruled out. But that isn’t the case here, where the analyses are just regression models include variables from different waves. Substantively, any number of third/confounding variables could affect both belief in conspiracy theories at one point in time and wellbeing at another point in time, and this study only has a strategy to control for two of them (turmoil, conspiracist ideation). The longitudinal analyses presented here therefore don’t provide much stronger evidence of causality than the cross-sectional ones.

What’s the solution to this? I should stress that I don’t think the answer is to back even further away from discussion of causal effects and present the paper as if it were only about associations. This paper is clearly motivated by questions about causal effects, and that’s totally ok. But I’d suggest really leaning into the topic of causal inferences, and how they can best be warranted and justified. Specifically:

1. Provide strong causal justifications for the variables controlled (and the ones you chose not to control). See Wysocki et al. (2022).

2. Be explicit about the fact that the paper is attempting to identify causal effects, and the assumptions required to interpret the estimates as unbiased estimates of causal effects (see Grosz et al., 2020; Rohrer, 2024).

3. Consider whether there are alternative data sources or analyses you could specify here that would provide stronger evidence of causal effects (see Rohrer, 2018). One possibility might be an additional longitudinal study, designed specifically for this paper, analysed via the random intercepts cross lagged panel model (Hamaker et al., 2015). Doing so might open up the opportunity to preregister analyses and subject the hypotheses to a severe test (see Lakens, 2019). Obviously this is a big ask in a peer review, and I’ll leave it up to the authors and editor to decide whether this is appropriate.

MINOR POINTS

4. The paper reports a lot of analyses. There are a bunch of different operationalisations of the dependent variables, different lags, models with each combination of the control variables, etc. Of course I would much rather the authors did that then cherrypick only a subset to report, but one problem resulting from the number of models and lack of a preregistration is that there are multiple reasonable interpretations of the results one could draw, considering they’re quite a mix of significant and non-significant coefficients for beliefs in conspiracy theories. The authors see a pattern that there is some effect of belief in conspiracy theories on wellbeing in the short term that disappears in the longer term (hence the non-significant effects at longer lags). This is one reasonable interpretation, but the differing findings at different lags could just as well be interpreted as suggesting there’s a fair bit of noise present and the evidence of effects is tentative and inconsistent. As a reader it’s consequently rather hard to tell to what extent the hypotheses faced a genuine risk of falsification. This is another reason why I think that another, preregistered, longitudinal study would really help here – albeit again I know this is a big ask.

5. P. 7, “Douglas and colleagues (2019) found that both belief in conspiracy theories regarding a given event or issue (e.g., 9/11 or genetically modified foods) and conspiracist ideation have primarily negative consequences”. The empirical findings cited in the paragraphs that follow here seem to be mainly correlational (‘consequences’ implies causal effects).

6. Economic turmoil is a key control variable in Study 1, but iss measured via just a single and relatively narrow item. If this item contains measurement error as a measure of turmoil (which will be the case), statistically controlling for it in a regression model doesn’t fully control for the effects of turmoil (see Westfall & Yarkoni, 2016). Similar issues, albeit less serious, apply to the other control measures.

7. Given the key role of the measure of stress as a dependent variable in Study 1, it’s likewise a bit surprising that it was measured via just a single ad hoc item. Why not use a validated measure of stress? The measurement error resulting from this decision won’t bias the results per se (since it’s measurement error in the DV), but will add noise and reduce power.

8. I’m not sure how much value there is in the factor analysis in of conspiracism items in Study 1. Given that the authors specified a 2-factor model without using a robust method for selecting the number of factors (c.f. the methods in Hayton et al., 2004; Zwick & Velicer, 1986), this can’t really be interpreted as a test of the idea that the factors “represent two related but distinct constructs.”

9. I take it the beta values reported are all standardized? It’d be good to make this clear at first use.

10. The interpretation that the “the benefits with respect to contentment that were evident at Wave 3 had dissipated substantially nine months later” isn’t strongly warranted by the findings. This claim implicitly suggests the effects were smaller at longer lags, but the models reported don’t test whether they were significantly smaller. The fact that they were mostly significant at short lags and not at long lags isn’t in of itself evidence of a significant difference in effects over time; “The difference between “significant” and “not significant” is not itself statistically significant” (Gelman & Stern, 2006). To test the idea that the effects dissipated over time would require an analysis specifically targeted to this question, likely involving some kind of interaction term.

11. I’m not convinced that “Coronavirus is actually no more dangerous than the common flu” is a conspiracy theory; it doesn’t allege a conspiracy. I can see the authors anticipated this objection and ran a small empirical study to show that people tend to think beliefs in this item would be associated with belief in conspiracy theories, but that’s not evidence of content validity per se. It might be useful to highlight in the main text how the main findings hold up when only the other, more clearly conspiratorial, item is treated as the dependent variable. (I think there might be some analyses of this in the supplementary materials but haven’t checked those in detail - apologies).

12. I don’t know that the use of factor scores for dependent variables in Study 2 was the optimal strategy. I appreciate why the authors did this, but the problem of factor score indeterminacy looms here (see Grice, 2001). There is an infinite number of different factor scores that can be computed from the same factor solution, and these different sets of factor scores would each produce different substantive findings. Would it not be a better idea to use structural equation modelling and treat the variables as latent?

13. Relatedly, the criteria on which the authors base the conclusion that the wave 3 wellbeing responses produced a 2-factor solution could have been clearer

14. Re. the control for conspiracist ideation, I can understand the authors’ rationale on this, but it’s hard to rule out the possibility that belief in specific conspiracy theories itself affects conspiracist ideation, which then goes on affect stress or wellbeing. Indeed, if I develop a belief in a specific conspiracy theory it’s not unreasonable to suppose this might make me more amenable to the idea that in general conspiracies happen frequently. If so, this would make conspiracist ideation a mediator and not just a confounding variable, such that controlling for it could add bias to the estimate of the effect of belief in specific theories. I don’t think this is a solvable problem per se, but do make the assumptions clear (e.g., to interpret the coefficients in the models with both controls as unbiased estimates of causal effects, we need to assume that belief in specific theories doesn’t affect either of the control variables).

15. “although our work suggests that believing a specific conspiracy theory may make a person feel better in the moment, these effects disappear over time, and have very real dangers for them and those around them.” I don’t think this study really shows evidence of dangers of beliefs in conspiracy theories; perhaps make it clearer that this claim stems from other evidence sources.

Again, despite my critiques I really liked this paper. It’s interesting, creative, and ambitious. With ambition comes tricky challenges to resolve, but that’s all part of the research process.

Best wishes,

Matt Williams

REFERENCES

Gelman, A., & Stern, H. (2006). The difference between “significant” and “not significant” is not itself statistically significant. The American Statistician, 60(4), 328–331. https://doi.org/10.1198/000313006X152649

Grice, J. W. (2001). Computing and evaluating factor scores. Psychological Methods, 6(4), 430–450.

Grosz, M. P., Rohrer, J. M., & Thoemmes, F. (2020). The taboo against explicit causal inference in nonexperimental psychology. Perspectives on Psychological Science, 15(5), 1243–1255. https://doi.org/10.1177/1745691620921521

Hamaker, E. L., Kuiper, R. M., & Grasman, R. P. P. P. (2015). A critique of the cross-lagged panel model. Psychological Methods, 20(1), 102–116. https://doi.org/10.1037/a0038889

Hayton, J. C., Allen, D. G., & Scarpello, V. (2004). Factor retention decisions in exploratory factor analysis: A tutorial on parallel analysis. Organizational Research Methods, 7(2), 191–205. https://doi.org/10.1177/1094428104263675

Lakens, D. (2019). The value of preregistration for psychological science: A conceptual analysis. Japanese Psychological Review, 62(3), 221–230. https://doi.org/10.24602/sjpr.62.3_221

Rohrer, J. M. (2018). Thinking clearly about correlations and causation: Graphical causal models for observational data. Advances in Methods and Practices in Psychological Science, 1(1), 27–42. https://doi.org/10.1177/2515245917745629

Rohrer, J. M. (2024). Causal inference for psychologists who think that causal inference is not for them. Social and Personality Psychology Compass, 18(3), e12948. https://doi.org/10.1111/spc3.12948

Westfall, J., & Yarkoni, T. (2016). Statistically controlling for confounding constructs is harder than you think. PLOS ONE, 11(3), e0152719. https://doi.org/10.1371/journal.pone.0152719

Wysocki, A. C., Lawson, K. M., & Rhemtulla, M. (2022). Statistical control requires causal justification. Advances in Methods and Practices in Psychological Science, 5(2), 25152459221095823. https://doi.org/10.1177/25152459221095823

Zwick, W. R., & Velicer, W. F. (1986). Comparison of five rules for determining the number of components to retain. Psychological Bulletin, 99(3), 432–442. https://doi.org/10.1037/0033-2909.99.3.432

Reviewer #3: Is there anything good about conspiracy beliefs? Belief in COVID-19 conspiracy theories can provide benefits to well-being

This is an interesting, thought provoking and nicely put together paper on an important topic. I enjoyed reading it. However, I do have one fairly major concern and will note a number of other areas where I think the paper can be strengthened. I’ll try to focus my review on providing requests for specific changes or additions, but I have to admit that my main concern doesn’t readily lend itself to a simple response, and the nature of any revision will depend on further examination of the data and an explication of the underlying causal model that the authors are using to make sense of the relationships between the constructs explored in the two studies.

1. Causal theory underpinning the aims and analyses.

At the core of the paper is the idea that specific conspiracy theory belief (CT) predicts improved mental health (MH) but only when we control for conspiracy ideation (CI) and/or economic turmoil (ET).

Across the two studies it is mostly found that, by itself, CT is associated with poorer MH or is not associated at all (consistent with previous research). When CI and ET are added as predictors (both by themselves and together) the relationship changes so that the estimate for CT is associated with improved MH. In other words, both CM and ET appear to be acting as suppressors. The question that needs answering is ‘why they are doing that?’ Is it possible that (1) the suppression is some kind of statistical artefact, or (2) does it suggest some deeper causal complexity that needs unpacking (e.g., a mediation model of some kind)?

My overriding question is ‘why do you want to control for CI and ET?’ - what kind of causal theory is motivating this? Why (and how), for instance, do we think that CI might be ‘hiding’ the true relationship between CT and MH? Both studies suggest that ‘that part of CT that is independent of CI and ET’ (call it CT-[CI+ET]) is positively related to MH. But what is CT-[CI+ET]? And what are the practical implications of this ‘thing’ being positively related to MH? At several points in the paper, it is implied that CT belief predicts better MH (as if CT-[CI+ET] is some kind of ‘purer measure’ of CT belief). Yet I don’t think this necessarily follows. Whatever CT-[CI+ET] is, it isn’t ‘CT simpliciter’. In fact, it seems to me to be a particularly strange kind of CT belief that is independent of the political and epistemic attitudes that underpin CI and independent of situational hardship drivers indexed by ET – it’s CT belief stripped of internal and situational mechanisms commonly associated with CT belief. It would be an understatement to say that I’m intrigued about the findings and would very much like to see some attempt to interpret the suppression effects.

Here are a few suggested additions that might make some of the relevant issues clearer to readers:

• To address statistical questions readers might have, include zero-order correlations between predictors and predictors and DVs/criterion variables and measures of multicollinearity (e.g., VIF). Simple slopes analyses or interaction analyses might also go some way to helping people unpack what is going on when the CI and ET are added to models.

• Re-work the introduction to make it clearer why the CI and ET variables are being controlled or are being treated as ‘confounds’ of the CT-MH relationship. If the causal relationships that underpin the decisions to do the analyses in the two studies could be represented in a Directed Acyclic Graph (DAG) I think readers will be better able to grasp the authors’ theoretical assumptions.

2. Clarifying what is meant by specific conspiracy theory and conspiracy ideation.

There’s a lack of clarity in the introduction about the difference between conspiracy ideation/mentality and belief in specific conspiracy theories. At times it seems like the term ‘specific conspiracy theory’ is intended to capture belief in a single specific CT. Yet in both studies ‘specific conspiracy theory’ is measured by the combined ratings of TWO specific CTs (although it’s good to see that the supplementary materials also examine ‘lone’ specific CTs).

On pages 4-5 it is suggested that Goertzel (1994) measured CI (“there was sufficient coherence in the data to create an index of conspiracist ideation”), yet he asked people about a collection of specific CTs (which isn’t the same thing as far as I can tell). The current paper seems to be using different definitions of specific CT and conspiracy ideation (where the average belief for a collection of specific conspiracy theories is considered to be a measure of CI) from that, that I’m used to. I don’t think this is what Imhoff et al. (2022), and the majority of papers in the 2024 special issue of Zeitschrift für Psychologie, suggest CI is. For instance, in Imhoff et al. (2022) Figure 1 represents specific conspiracy belief as mean level endorsement of a collection of specific CTs (where the number of CTs ranges from from N=99, N=6, N=2). Similarly, Trella et al. (2024) note that “the construct we might call the conspiracy mentality is captured by the CMQ while the construct we might call belief in conspiracy theories is captured by aggregate scales such as the BCTI.” (my emphasis). The BCTI is consists of 15 specific CTs.

3. Measures of conspiracy ideation used in the two studies.

Study 2 measures CI using the CMQ, and Study 1 uses the GCBS. Although, it’s fair to say that the latter has been used as a conspiracy mentality measure by other authors, I think it is important to acknowledge that GCBS doesn’t measure exactly the same thing at the CMQ (and perhaps that might explain some of the differences in the findings across the two studies).

The items in the CMQ (and similar scales like the CMS) aren’t explicitly about conspiracies at all but about beliefs related to conspiracy belief, although not necessarily so. This probably explains why the distributions of scores on the CMQ and CMS differ from those for specific conspiracy theory belief (see Figure 1 of Imhoff et al., 2022).

If you look at the CMQ items they attempt to tap a variety of beliefs about authorities and political processes (such as powerful groups keeping secrets, things not being as they seem, coincidences hiding deeper truths and so on) whereas the GCBS measures beliefs about ‘high-level’ conspiracy theories that do not include detail about specific protagonists or events. The two types of measure are not abstract (an idea that is canvassed on p. 6 of the manuscript) in the same way. Sutton et al. (2024) claim that conspiracy mentality refers to an abstract, generalized political attitude. By contrast the GCBS is abstract in the sense that it measures beliefs about types or clusters of conspiracy theories.

4. Measures of specific conspiracy theories used in the two studies.

The items used to measure specific conspiracy theories in Study 2 aren’t, strictly speaking, descriptions of specific conspiracy theories (although they are certainly ‘conspiracy-adjacent’). Neither of these items include explicit mention of a conspiracy or identify who the conspirators are or that the activity has been carried out for malicious reasons. I think it is worth pointing out this difference between the two studies. If you can find evidence to support the idea that people interpret statements like those used in Study 2 as conspiracy theory statements, that would be worth including.

List of Recommended Changes

1. Articulate what you think ‘CT belief controlling for CI and ET’ is (conceptually/theoretically-speaking) around pp. 9-10. DAGs might be useful.

2. Consider examining interactions or simple slopes to better understand how CT belief varies with CI (and ET).

3. Related to (1), ensure that claims about mental health benefits is are clearly associated with ‘CT belief controlling for CM and ET’ and not CT belief simpliciter.

4. Clarify what you mean by belief in specific conspiracy theories and conspiracy ideation. Ensure it is consistent with existing ideas (e.g., Imhoff et al., 2022) or that it is clear that you have a different take on this distinction from other authors.

5. Related to (4), clarify what you mean by conspiracy ideation/mentality.

6. Provide VIF/multicollinearity information and more detail about the standardisation procedure used on p. 15. I presume this means that z-scores were calculated rather than variables being mean-centred? Was this done to manage multicollinearity?

7. Provide zero-order correlation information.

8. Clarify paragraph 1 on p. 10. I don’t understand “may fare better than those left with an explanation whose scope seems to not match the large scale of the issue”. What is this alternative (non-conspiracy?) explanation alluded to here? Why is ‘scope matching’ an issue? Why doesn’t conspiracy ideation provide adequate ‘scope’? And how does the claim in the final sentence relate to the rest of the paragraph? I found this paragraph particularly difficult to make sense of and, given it is part of the justification for ‘controlling for’ CI, seems to be a key part of the rationale for the overall analytic approach taken in the paper.

9. p. 16 – I’d be inclined to avoid talking about marginal significance – I don’t think there it makes a great deal of difference it you just say that there’s no significant relationship between conspiracy ideation and stress for Study 1.

10. Similarly p = .09 is called marginally significant on p. 28. I’d be inclined to simply say that CT + CM does not predict negative symptomology. Perhaps use the same approach to describe this kind of finding as has been done for the p =.15 finding at the bottom on p. 28.

11. Note that GCBS and CMQ don’t measure exactly the same thing although they seem to correlate reasonably well (e.g., in Swami et al., 2017) – as does pretty much everything in the conspiracy world it seems.

12. Note that the ‘conspiracy-adjacent items’ in Study 2 aren’t explicit specific conspiracy theories (like those used in Study 1). They don’t describe a conspiracy or conspirators that is done for malicious reasons.

13. The timing of the measures in Study 2 raises some questions. I’m not sure if it sufficiently clear that the Wave 3 analyses use ‘past measures’ of economic turmoil, ideation, and CT belief that are 3-4 months older than the Wave 3 subjective wellbeing measure. That is, neither the Wave 3 or the Wave 5 analyses examine current levels of economic turmoil, ideation, and CT belief. We simply don’t know if the Wave 1 and Wave 2 measures are a reasonable measure of current economic turmoil, ideation, and CT belief for the analyses.

6. PLOS authors have the option to publish the peer review history of their article (what does this mean? ). If published, this will include your full peer review and any attached files.

**Do you want your identity to be public for this peer review?** For information about this choice, including consent withdrawal, please see our Privacy Policy .

Reviewer #1: No

Reviewer #2: **Yes: ** Matt Williams

Reviewer #3: No

---

## [Author Response · Author response to Decision Letter 0]

5 Sep 2024

See the uploaded Response to Reviewers document.

---

## [Decision Letter · Decision Letter 1]

18 Oct 2024

PONE-D-24-11501R1Is there anything good about conspiracy beliefs? Belief in COVID-19 conspiracy theories can provide benefits to well-beingPLOS ONE

Dear Dr. Fazio,

Thank you for submitting your manuscript to PLOS ONE. After careful consideration, we feel that it has merit but does not fully meet PLOS ONE’s publication criteria as it currently stands. Therefore, we invite you to submit a revised version of the manuscript that addresses the points raised during the review process. Overall, all three reviewers acknowledge the improvements made to the manuscript and appreciate the revisions. However, since two reviewers recommend minor revisions, particularly concerning the conceptualization of CTs, I kindly ask that you further refine the relevant sections and address their comments.  

We look forward to receiving your revised manuscript.

Kind regards,

Cengiz Erisen

Academic Editor

PLOS ONE

Journal Requirements:

Reviewers' comments:

Reviewer's Responses to Questions

**Comments to the Author**

1. If the authors have adequately addressed your comments raised in a previous round of review and you feel that this manuscript is now acceptable for publication, you may indicate that here to bypass the “Comments to the Author” section, enter your conflict of interest statement in the “Confidential to Editor” section, and submit your "Accept" recommendation.

Reviewer #1: All comments have been addressed

Reviewer #2: (No Response)

Reviewer #3: (No Response)

2. Is the manuscript technically sound, and do the data support the conclusions?

Reviewer #1: Yes

Reviewer #2: Partly

Reviewer #3: Partly

3. Has the statistical analysis been performed appropriately and rigorously? 

Reviewer #1: Yes

Reviewer #2: Yes

Reviewer #3: Yes

4. Have the authors made all data underlying the findings in their manuscript fully available?

Reviewer #1: Yes

Reviewer #2: Yes

Reviewer #3: Yes

5. Is the manuscript presented in an intelligible fashion and written in standard English?

Reviewer #1: Yes

Reviewer #2: Yes

Reviewer #3: Yes

6. Review Comments to the Author

Reviewer #1: (No Response)

Reviewer #2: My thanks for the thorough and extensive work you've done in response to the reviews, and for your clear responses. I do think this manuscript is greatly improved. At this stage I have a small number of remaining points.

My major criticism of the original manuscript was in relation to the issue of causal inferences. In response to this, you have provided causal justifications for the control variables, been explicit about the intent to draw causal inferences, and considered additional sources of evidence for causal effects you could provide. On the latter point you were able to provide a propensity scores analysis. These are all great improvements. What I'd like to see now is a re-calibration of specific conclusions made in the title, abstract, and main body to ensure that they are calibrated to the evidence for causal effects (which is only tentative), and to be consistent with the (accurate) caveat that "strong causal claims are unwarranted". I apologise for not making this expectation explicit in my initial review. Some of the specific claims I noticed that still make causal inferences without highlighting uncertainties include:

1. "Belief in COVID-19 conspiracy theories can provide benefits to well-being". (It's a really nice title, but in this form it just isn't warranted by the evidence in the paper)

2. "The current research provides evidence for benefits of an event conspiracy belief" (abstract)

3. "conspiracy beliefs do provide at least temporary intrapersonal benefits" (abstract)

4. "To our knowledge, this is the first instance in which a beneficial intrapersonal effect of believing conspiracy theories on stress has been documented." (p. 22)

5. "Most importantly, this study [2] provides evidence that believing in specific conspiracy theories can have benefits for the believer—conceptually replicating the results of Study 1" (p. 39).

6. "Putting these findings together suggests that the benefits of believing specific conspiracy theories operate by boosting positive affect" (p. 40-41).

There could be others I've missed. I know that there's some great nuanced thinking about causality in the response letter, and in some other parts of the manuscript, and I'm definitely not suggesting you just delete all these sentences. But considered in isolation these are unambiguous "strong causal claims" that don't acknowledge uncertainty. Inevitably, it will be these strong causal claims that will be picked up and amplified by journalists and researchers citing your work. So please present *every* causal inference in such a way that it acknowledges relevant uncertainties and/or assumptions you're making.

7. In addition, I'd like to see that paragraph about causal limitations in the discussion section expanded into something a bit more comprehensive. What assumptions do readers need to make to interpret your beta values as unbiased estimates of causal effects? How realistic are those assumptions? What would happen if they're breached? E.g., one such assumption is that aside from conspiracist ideation and turmoil there exists no other third variable that affects both belief in specific conspiracy theories and wellbeing...

8. In my first review I pointed out that the pattern of findings across studies was consistent with your interpretation, but also consistent with an interpretation that the evidence of effects is tentative and inconsistent. I really appreciate your response but I'm afraid I remain unconvinced! In Study 2 there are 4 wellbeing outcomes (negative symptomology and contentment, both at waves 3 and 5). Only one of these 4 is significantly & positively predicted by belief in conspiracy theories (contentment at wave 3). And this finding isn't quite consistent with Study 1, because Study 1 used stress as an outcome measure, which is negative symptomology, not contentment. The positive coefficient in Study 2 is occurring only in the (unusual) circumstance of a bivariate positive correlation between belief in CTs and contentment, which isn't very consistent with the "suppressor variables" idea articulated earlier in the paper. As it stands, this paper presents only tentative evidence for positive effects of belief in CTs (both due to the causal inference issue and the not-entirely-consistent results). That *doesn't* mean it's not a useful contribution to the literature, and I'm 100% ok with the fact that you have decided an additional study isn't feasible... but I'd like to see that discussion section show a bit more acknowledgment of how much uncertainty remains until more followup work is done.

9. Re. my original point 6, I think you make good points about the merits of single items. However, the fact remains that your control variables all inevitably have at least *some* measurement error attached, and your regression models assume they don't. Even without any other confounds this would be enough to cause the estimates of the focal causal effects to be biased (see Westfall & Yarkoni, 2016). I do think you need to acknowledge this in the manuscript.

Presentational points:

10. Figure 1 is a very useful addition. The arrows ppointing to wellbeing don't render clearly in the pdf, so it'd be worth double-checking they look ok during copyediting.

11. P. 29, the 2 in the chi-square symbols should be superscript.

Reviewer #3: The authors have done a nice job attending to many of the comments from the reviewers. There are two areas that I think could still do with some attention:

1. Conspiracy item used in Study 2.

“COVID-19 is no more dangerous than the common cold" isn’t a conspiracy theory item - at best we can call it an empirically unsupported claim that underpins a number of conspiracy theories. I think it would be appropriate for the authors to describe it thus. The survey (described on p. 26 of the manuscript) asking people whether they think someone who endorses the claim believes a conspiracy doesn’t alter this fact. I suspect quite a few non-conspiracy claims might result in a similar type of response (e.g., “January 6th protestors had every right to enter the Capitol” or even “Trump tells the truth” or “There are some things that people see in the sky that aren’t easily explainable”). Endorsing such claims might be something you’d expect of someone who believes particular conspiracies but that doesn’t make the claims themselves conspiracy theories and, more importantly, it isn’t strong evidence that a participant who endorses them actually believes a particular conspiracy. Moreover, the claim doesn’t even ‘pick out’ a single conspiracy theory - it is consistent with a variety of conspiracies as well a number of non-conspiracy beliefs - someone could believe this claim without thinking that it involves secrecy or coverups, or that a group has malevolent intentions.

2. Clarify the distinction between conspiracy ideation (CI) and (event) conspiracy theory belief (CT) that is central to the claims of the paper.

My comments in the initial review were intended to be about the conceptual distinction being made (or implied) in the research, rather than a point about the interpretation of analyses - however, I can see why they were interpreted that way. Apologies for my lack clarity. I should also have made it clearer that my main concern is with the idea that ‘CT controlling for CI’ makes good conceptual sense. (the economic turmoil variable seems more theoretically tractable to me). From a statistical point of view the paper seems fine to me (although I agree with Reviewer 2’s points about the need to be cautious about drawing causal conclusions from observational/correlational analyses). The authors provide evidence for CI and CT being statistically different constructs (using factor analysis) despite them being quite strongly correlated (something that has been observed before of course - Imhoff et al., 2022 explore some possible reasons why this is the case). The majority of the regression analyses show that the CI measure acts as a suppressor on CT measure-wellbeing relationships.

So, to repeat, my worry is that there’s still a lack of conceptual clarity about how we should make sense of these findings. I think that the discussion on pages 10-13 of the new manuscript goes a long way towards providing the kind of substantial theoretical rationale that is needed. The authors seem to see the CI-CT distinction differently from authors like Imhoff do, where CI is understood as a mechanism (disposition, propensity, “distal predictor”) that “largely determines” (in the words of Imhoff & Bruder, 2014) belief in specific CTs. Framed this way it doesn’t make much sense (to me) to include both CI (mechanism/part cause) and CT (effect) in a model of predictors of wellbeing. It is difficult to know how we can interpret CI as a suppressor under this reading.

By contrast with this ‘mechanism-effect view’, the authors argue that CI is a construct associated with chronic, maladaptive rumination whereas specific/event CT belief is an adaptive attempt at explanation that makes sense of troubling phenomenon in a satisfying fashion. These differences explain why wellbeing is related to the two constructs in different ways. This is an interesting idea but I do wonder whether it involves a novel redefinition of CI (and CT for that matter - now we are treating event conspiracy beliefs as something different from how other researchers talk about specific conspiracy beliefs) in a slightly question begging way - that is, the constructs get redefined in order to justify the analyses.

The discussion on p. 13 of the edited manuscript speculating about why a conspiracy explanation (event conspiracy belief) might be less stressful than a non-conspiratorial alternative (that posits an unpredictable world and/or a cause that is not in proportion with the effect) is intriguing (although it does seem at odds with the findings of Stojanov & Halberstadt, 2020 who found little evidence that perceived lack of control predicted conspiracy belief).

The following paragraph (lines 17-20) that speculates about how CI might result in lower wellbeing/higher stress also sounds plausible - yet together, the two do seem at odds with each other. That is, CI predicts poorer wellbeing because it’s stressful to ruminate about dark forces affecting ones life - but, for some reason, ruminating about specific dark forces affecting a specific event in ones life has the opposite effect.

Could we not (a priori) flip the arguments around and argue that CI should be LESS stressful than the alternative (having a propensity to think the world is unpredictable and often big things happen because of small causes)? And could we not also argue that event CT belief should be MORE stressful (than other kinds of explanations) because it is stressful to believe that dark forces are affecting ones life? It seems to me that, the proposed explanation of the different relationships CI and CT have with wellbeing, needs to be bolstered by, for instance, ruling out the possibility that CI could be MORE satisfying and LESS stressful than alternative explanatory propensities.

References mentioned

Imhoff, R., & Bruder, M. (2014). Speaking (un–)truth to power: Conspiracy mentality as a generalised political attitude. European Journal of Personality, 28(1), 25–43. https://doi.org/10.1002/per.1930

Stojanov, A., & Halberstadt, J. (2020). Does lack of control lead to conspiracy beliefs? A meta‐analysis. European Journal of Social Psychology, 50(5), 955–968. https://doi.org/10.1002/ejsp.2690

7. PLOS authors have the option to publish the peer review history of their article (what does this mean? ). If published, this will include your full peer review and any attached files.

**Do you want your identity to be public for this peer review?** For information about this choice, including consent withdrawal, please see our Privacy Policy .

Reviewer #1: No

Reviewer #2: No

Reviewer #3: No

---

## [Author Response · Author response to Decision Letter 1]

24 Dec 2024

See the uploaded document for a more readable version.

Response to reviewers

We would like to thank the editor and reviewers for their favorable reactions to our manuscript and for taking the time to provide helpful comments. We have carefully considered each comment, and have made multiple revisions to the manuscript. As was the case with the earlier submission, our manuscript has greatly benefited from the feedback. We reproduce each comment offered in the action letter below in blue font, and provide responses in this black font. All page numbers referenced correspond to the “Manuscript with Track Changes” document.

Editor:

Overall, all three reviewers acknowledge the improvements made to the manuscript and appreciate the revisions. However, since two reviewers recommend minor revisions, particularly concerning the conceptualization of CTs, I kindly ask that you further refine the relevant sections and address their comments.

We were glad to hear that our revision was well-received. We provide in-depth responses to the reviewer comments in the pages below. 

Reviewer #2:

My major criticism of the original manuscript was in relation to the issue of causal inferences. In response to this, you have provided causal justifications for the control variables, been explicit about the intent to draw causal inferences, and considered additional sources of evidence for causal effects you could provide. On the latter point you were able to provide a propensity scores analysis. These are all great improvements. What I'd like to see now is a re-calibration of specific conclusions made in the title, abstract, and main body to ensure that they are calibrated to the evidence for causal effects (which is only tentative), and to be consistent with the (accurate) caveat that "strong causal claims are unwarranted". I apologise for not making this expectation explicit in my initial review. Some of the specific claims I noticed that still make causal inferences without highlighting uncertainties include:

Thank you for bringing attention to this issue. Points 1-6 below reference different quotes that can be recalibrated to better communicate the non-experimental nature of the data. For each point, we provide the edited text in its broader context and its associated page number.

1. "Belief in COVID-19 conspiracy theories can provide benefits to well-being". (It's a really nice title, but in this form it just isn't warranted by the evidence in the paper)

“Is there anything good about conspiracy beliefs? Belief in COVID-19 conspiracy theories is associated with benefits to well-being” (p. 1)

2. "The current research provides evidence for benefits of an event conspiracy belief" (abstract)

“The current research provides correlational evidence for a link between well-being and an event conspiracy belief by teasing apart its effect from (1) the influence of experiencing turmoil that nudges people toward believing the event conspiracy theory in the first place and (2) conspiracist ideation—the general tendency to engage in conspiratorial thinking.” (p. 2)

3. "conspiracy beliefs do provide at least temporary intrapersonal benefits" (abstract)

“These findings suggest that despite their numerous negative consequences, event conspiracy beliefs are associated with at least temporary intrapersonal benefits.” (p. 2)

4. "To our knowledge, this is the first instance in which a beneficial intrapersonal effect of believing conspiracy theories on stress has been documented." (p. 22)

“To our knowledge, this is the first instance in which a positive relation between believing conspiracy theories and stress has been documented.” (p. 22)

5. "Most importantly, this study [2] provides evidence that believing in specific conspiracy theories can have benefits for the believer—conceptually replicating the results of Study 1" (p. 39).

“Most importantly, this study provides evidence that believing in specific conspiracy theories can be associated with boosts in well-being—conceptually replicating the results of Study 1—and clarifies that such relations are specific to increased contentment for the believer.” (p. 40)

6. "Putting these findings together suggests that the benefits of believing specific conspiracy theories operate by boosting positive affect" (p. 40-41).

“Putting these findings together suggests that any association between believing specific conspiracy theories and well-being is restricted to positive affect—which may make people feel more capable of dealing with the threats they face—but only for a limited amount of time.” (p. 42)

There could be others I've missed. I know that there's some great nuanced thinking about causality in the response letter, and in some other parts of the manuscript, and I'm definitely not suggesting you just delete all these sentences. But considered in isolation these are unambiguous "strong causal claims" that don't acknowledge uncertainty. Inevitably, it will be these strong causal claims that will be picked up and amplified by journalists and researchers citing your work. So please present *every* causal inference in such a way that it acknowledges relevant uncertainties and/or assumptions you're making.

We reviewed the manuscript thoroughly to find other instances in which we may be able to recalibrate the claims. We list such instances below:

Original quote: “However, the contentment benefit conferred by an event conspiracy belief recedes over time.”

Revised quote: “However, the relation between COVID-19 conspiracy belief and contentment diminishes in size over time.” (p. 2)

Original quote: “In summary, we propose that believing conspiracy theories may indeed have benefits for the believer, but that uncovering these benefits requires disentangling the influence of endorsing specific conspiracy theories from…”

Revised quote: “In summary, we propose that believing conspiracy theories may indeed be associated with benefits for the believer, but that uncovering these associations requires disentangling the influence of endorsing specific conspiracy theories from…” (p. 14)

Original quote: “In Study 2, we extended our prior findings by examining how belief in COVID-19 conspiracy theories influences different aspects of well-being and how these relations change over time.”

Revised quote: “In Study 2, we extended our prior findings by examining how belief in COVID-19 conspiracy theories relates to different aspects of well-being and how these relations change over time.” (p. 38)

7. In addition, I'd like to see that paragraph about causal limitations in the discussion section expanded into something a bit more comprehensive. What assumptions do readers need to make to interpret your beta values as unbiased estimates of causal effects? How realistic are those assumptions? What would happen if they're breached? E.g., one such assumption is that aside from conspiracist ideation and turmoil there exists no other third variable that affects both belief in specific conspiracy theories and wellbeing...

Thank you for encouraging us to expand on the conditions necessary to interpret our results as unbiased estimates of causal effects. We have taken this opportunity to go beyond the suggestion and discuss the criteria for establishing causality more generally. We follow this exposition by describing how our data fit with these criteria:

Establishing causality requires demonstrating evidence of covariation, temporal precedence, as well as ruling out plausible alternative explanations (Shadish et al., 2002). Across two studies, we provide evidence of covariation between event conspiracy beliefs and well-being. Although not definitive, the prospective nature of Study 2 provides some evidence regarding temporal precedence of the proposed relation between event conspiracy beliefs and well-being. Lastly, interpreting our results as unbiased estimates of the causal effect of event conspiracy beliefs on well-being requires ruling out potential third variables. Naturally, a failure to rule out potential third variables would bias the estimates of the relation between event conspiracy beliefs and well-being, increasing the possibility of a spurious correlation. Although this research cannot definitively rule out all alternative explanations, it does leverage multiple regression and propensity score weighting analyses to rule out the most relevant situational, dispositional, and demographic variables as competing explanations based on the current state of the literature. In so doing, we furnish evidence that is consistent with the idea that believing specific conspiracy theories can have benefits for well-being. While it is possible that confounders other than those tested exist, we are unaware of likely candidates for the role. Nevertheless, it should be kept in mind that the current findings represent merely a step, albeit an important one in our view, in building a case for a causal relation. The research presented here establishes a positive association between event conspiracy beliefs and well-being. (pgs. 44)

8. In my first review I pointed out that the pattern of findings across studies was consistent with your interpretation, but also consistent with an interpretation that the evidence of effects is tentative and inconsistent. I really appreciate your response but I'm afraid I remain unconvinced! In Study 2 there are 4 wellbeing outcomes (negative symptomology and contentment, both at waves 3 and 5). Only one of these 4 is significantly & positively predicted by belief in conspiracy theories (contentment at wave 3). And this finding isn't quite consistent with Study 1, because Study 1 used stress as an outcome measure, which is negative symptomology, not contentment. The positive coefficient in Study 2 is occurring only in the (unusual) circumstance of a bivariate positive correlation between belief in CTs and contentment, which isn't very consistent with the "suppressor variables" idea articulated earlier in the paper. As it stands, this paper presents only tentative evidence for positive effects of belief in CTs (both due to the causal inference issue and the not-entirely-consistent results). That *doesn't* mean it's not a useful contribution to the literature, and I'm 100% ok with the fact that you have decided an additional study isn't feasible... but I'd like to see that discussion section show a bit more acknowledgment of how much uncertainty remains until more followup work is done.

We appreciate the encouragement to better communicate the nature of our results. First, we would like to clarify that stress is not necessarily considered negative symptomology. Although we can certainly understand the link being made in the comment given the uncomfortable state that characterizes feeling stressed, the stress construct is recognized as being related to but distinct from well-being by the leading figures in that area of research. As indicated in the original revision, stress is used as a proxy for well-being in Study 1. To ensure that there is no confusion about the nature of the stress measure, we have added a sentence clarifying the nature of the construct and its relation to well-being:

As a proxy for well-being, we assessed people’s reports regarding the general level of stress they were currently experiencing. Stress refers to an individual’s appraisal of the demands placed on them in relation to their resources (Moskowitz, 2007), and is related to but distinct from well-being (Ng & Diener, 2022). (pg. 15)

That said, we do see value in better communicating the alternative interpretation of our results articulated above and the need for independent validation. We have inserted text into page 42 to communicate these points:

We interpret the results as providing robust evidence for our hypothesis of a relation between event conspiracy beliefs and well-being. However, the nature of the results in Study 2—in which COVID-19 conspiracy beliefs selectively predict contentment in the short-term but not the long-term—may also be interpreted as indicating that the relation between event conspiracy beliefs and well-being is at least somewhat tentative. As with all research documenting novel relations, independent replication is required.

9. Re. my original point 6, I think you make good points about the merits of single items. However, the fact remains that your control variables all inevitably have at least *some* measurement error attached, and your regression models assume they don't. Even without any other confounds this would be enough to cause the estimates of the focal causal effects to be biased (see Westfall & Yarkoni, 2016). I do think you need to acknowledge this in the manuscript.

We thank the reviewer for bringing attention to this issue. Before directly addressing the key point of this comment, we would like to point out that we use Structural Equation Modeling to ensure that the multiple regression analyses reported in the manuscript are robust to tools that take into account measurement error for multi-item measures, which we report in the Supplemental Material. Of course, single-item measures are treated as observed variables in SEM, and thus, the concern regarding the measurement error associated with them is not addressed by these models. We would like to note, however, that when using a large sample like that of Study 2, the introduction of measurement error almost exclusively reduces the effect size of the relation in question: “In the large-N scenario, adding measurement error will almost always reduce the observed correlation between x and y (see the figure, left panel).” (p. 585; Loken & Gelman, 2017). That said, we do see value in directing future research to circumvent such issues by using multi-item measures. We have inserted text on page 42 to encourage future replication attempts to use multi-item measures to assess economic turmoil:

Despite finding robust associations between economic turmoil and well-being, one valuable aspect of a future replication attempt would be the measurement of economic turmoil using a multi-item measure to ensure that this construct is assessed more comprehensively and measurement error can be modeled.

Presentational points:

10. Figure 1 is a very useful addition. The arrows ppointing to wellbeing don't render clearly in the pdf, so it'd be worth double-checking they look ok during copyediting.

Thank you for bringing our attention to this potential issue. We will attach the image file to our submission to ensure that the image is rendered in high quality.

11. P. 29, the 2 in the chi-square symbols should be superscript.

Thank you for finding these errors. The 2 in the chi-square symbol has been changed into a superscript. 

Reviewer #3:

The authors have done a nice job attending to many of the comments from the reviewers. There are two areas that I think could still do with some attention:

1. Conspiracy item used in Study 2.

“COVID-19 is no more dangerous than the common cold" isn’t a conspiracy theory item - at best we can call it an empirically unsupported claim that underpins a number of conspiracy theories. I think it would be appropriate for the authors to describe it thus. The survey (described on p. 26 of the manuscript) asking people whether they think someone who endorses the claim believes a conspiracy doesn’t alter this fact. I suspect quite a few non-conspiracy claims might result in a similar type of response (e.g., “January 6th protestors had every right to enter the Capitol” or even “Trump tells the truth” or “There are some things that people see in the sky that aren’t easily explainable”). Endorsing such claims might be something you’d expect of someone who believes particular conspiracies but that doesn’t make the claims themselves conspiracy theories and, more importantly, it isn’t strong evidence that a participant who endorses them actually believes a particular conspiracy. Moreover, the claim doesn’t even ‘pick out’ a single conspiracy theory - it is consistent with a variety of conspiracies as well a number of non-conspiracy beliefs - someone could believe this claim without thinking that it involves secrecy or coverups, or that a group has malevolent intentions.

We appreciate the opportunity to be more precise about the nature of t

---

## [Decision Letter · Decision Letter 2]

23 Jan 2025

PONE-D-24-11501R2Is there anything good about conspiracy beliefs? Belief in COVID-19 conspiracy theories is associated with benefits to well-beingPLOS ONE

Dear Dr. Fazio,

Thank you for submitting your manuscript to PLOS ONE. After careful consideration, we feel that your manuscript could be published but at this point does not fully satisfy R2. I believe these comments could be quickly addressed in a final revision after which I will make an in-house decision. 

We look forward to receiving your revised manuscript.

Kind regards,

Cengiz Erisen

Academic Editor

PLOS ONE

Journal Requirements:

Reviewers' comments:

Reviewer's Responses to Questions

**Comments to the Author**

1. If the authors have adequately addressed your comments raised in a previous round of review and you feel that this manuscript is now acceptable for publication, you may indicate that here to bypass the “Comments to the Author” section, enter your conflict of interest statement in the “Confidential to Editor” section, and submit your "Accept" recommendation.

Reviewer #2: (No Response)

2. Is the manuscript technically sound, and do the data support the conclusions?

Reviewer #2: Partly

3. Has the statistical analysis been performed appropriately and rigorously? 

Reviewer #2: Yes

4. Have the authors made all data underlying the findings in their manuscript fully available?

Reviewer #2: Yes

5. Is the manuscript presented in an intelligible fashion and written in standard English?

Reviewer #2: Yes

6. Review Comments to the Author

Reviewer #2: Given that this is my third review of this manuscript, I have not reviewed it in its entirety, instead focusing on the authors’ revisions in relation to my points. At this point I was really hoping to say nothing more than “It’s great, accept it!” However, not quite all the points I raised have been addressed yet. Specifically:

1. Some of the causal claims have been modified in an appropriate fashion. However, in relation to the excerpts I highlighted in points 2, 3 and 5, the authors are still making causal inferences (“effect”, “benefits”, “boosts”), and without acknowledging the relevant uncertainties and/or assumptions. Even little cross-references like “but see the limitations subsection for the assumptions required for this interpretation…” could have done the trick.

2. In my point 7, I asked for the “paragraph about causal limitations in the discussion section expanded into something a bit more comprehensive. What assumptions do readers need to make to interpret your beta values as unbiased estimates of causal effects? How realistic are those assumptions? What would happen if they're breached? E.g., one such assumption is that aside from conspiracist ideation and turmoil there exists no other third variable that affects both belief in specific conspiracy theories and wellbeing...” The authors have improved this paragraph, but not listed all these assumptions (just one that I happened to use as an example). What about the assumptions that the control variables are not colliders or mediators? To help them produce a more sophisticated description I’d suggest reading contemporary literature on causal inference (e.g. Rohrer, 2018, 2024). Thinking about simple concepts like covariance, temporal precedence and excluding alternative explanations is helpful when introducing students to causal inference, but at this level a bit more depth and specificity is called for.

3. Regarding my point 6: “the fact remains that your control variables all inevitably have at least *some* measurement error attached, your regression models assume they don't. Even without any other confounds this would be enough to cause the estimates of the focal causal effects to be biased (see Westfall & Yarkoni, 2016). I do think this needs to be acknowledged in the manuscript.” The authors respond by pointing to simulation by Gelman and Loken (2017). That simulation is interesting but of limited relevance to the current study: It’s a simulation relating to a bivariate relationship, not one where with control variables measured with error. Westfall and Yarkoni show that measurement error in covariates can cause biased estimates, and large sample sizes do not resolve this problem (in some cases they can even make it worse). The authors thus haven’t yet addressed my request.

I would gently suggest to the authors that when a peer reviewer or editor offers an R&R decision, that that you carefully and thoroughly perform what they ask before resubmitting. I know it can be hard to modify aspects of a manuscript that you really like, but multiple rounds of review aren’t good for any of us. I like this study, and with a bit more proactiveness in the R&Rs it could have been accepted some time ago. I look forward to seeing it published.

REFERENCES

Rohrer, J. M. (2018). Thinking clearly about correlations and causation: Graphical causal models for observational data. Advances in Methods and Practices in Psychological Science, 1(1), 27–42. https://doi.org/10.1177/2515245917745629

Rohrer, J. M. (2024). Causal inference for psychologists who think that causal inference is not for them. Social and Personality Psychology Compass, 18(3), e12948. https://doi.org/10.1111/spc3.12948

Westfall, J., & Yarkoni, T. (2016). Statistically controlling for confounding constructs is harder than you think. PLOS ONE, 11(3), e0152719. https://doi.org/10.1371/journal.pone.0152719

7. PLOS authors have the option to publish the peer review history of their article (what does this mean? ). If published, this will include your full peer review and any attached files.

**Do you want your identity to be public for this peer review?** For information about this choice, including consent withdrawal, please see our Privacy Policy .

Reviewer #2: No

---

## [Author Response · Author response to Decision Letter 2]

29 Jan 2025

Please see the document labeled "Response to Reviewers_Jan29."

---

## [Editor Report · Decision Letter 3]

11 Feb 2025

Is there anything good about conspiracy beliefs? Belief in COVID-19 conspiracy theories is associated with benefits to well-being

PONE-D-24-11501R3

Dear Dr. Fazio,

Thank you very much for your kind interest in making the final revisions to your manuscript. Based on my reading of the final version, I am pleased to inform you that your manuscript has been judged scientifically suitable for publication. Congratulations!

Kind regards,

Cengiz Erisen

Academic Editor

PLOS ONE
---

## [Editor Report · Acceptance letter]

PONE-D-24-11501R3

PLOS ONE

Dear Dr. Fazio,

I'm pleased to inform you that your manuscript has been deemed suitable for publication in PLOS ONE. Congratulations! Your manuscript is now being handed over to our production team.

Kind regards,

on behalf of

Dr. Cengiz Erisen

Academic Editor

PLOS ONE